

# Hole-induced anomaly in the thermodynamic behavior of a one-dimensional Bose gas

Giulia De Rosi[1⋆], Riccardo Rota[2], Grigori E. Astrakharchik[1] and Jordi Boronat[1]

**1** Departament de Física, Universitat Politècnica de Catalunya,
Campus Nord B4-B5, 08034 Barcelona, Spain
**2** Institute of Physics, Ecole Polytechnique Fédérale de Lausanne (EPFL),
CH-1015 Lausanne, Switzerland

⋆ giulia.de.rosi@upc.edu

## Abstract

We reveal an intriguing anomaly in the temperature dependence of the specific heat of a one-dimensional Bose gas. The observed peak holds for arbitrary interaction and remembers a superfluid-to-normal phase transition in higher dimensions, but phase transitions are not allowed in one dimension. The presence of the anomaly signals a region of unpopulated states which behaves as an energy gap and is located below the hole branch in the excitation spectrum. The anomaly temperature is found to be of the same order of the energy of the maximum of the hole branch. We rely on the Bethe Ansatz to obtain the specific heat exactly and provide interpretations of the analytically tractable limits. The dynamic structure factor is computed with the Path Integral Monte Carlo method for the first time. We notice that at temperatures similar to the anomaly threshold, the energy of the thermal fluctuations become comparable with the maximal hole energy, leading to a qualitative change in the structure of excitations. This excitation pattern experiences the breakdown of the quasiparticle description for any value of the interaction strength at the anomaly, similarly to any superfluid phase transition at the critical temperature. We provide indications for future observations and how the hole anomaly can be employed for in-situ thermometry, identifying different collisional regimes and understanding other anomalies in atomic, solid-state, electronic, spin-chain and ladder systems.

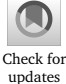

# 1   Introduction

In a number of classical and quantum many-body systems, the specific heat as a function of temperature shows a thermal feature, often referred to as an *Anomaly* and which occurs for different reasons. A first example is provided by the onset of a thermal second-order phase transition where the specific heat shows a sharp peak located at the critical temperature [1]. This anomaly appears in very different transitions: normal/superconductor [2], normal/superfluid in helium [3,4] and in strongly-interacting ultracold atomic Fermi gases [5,6]. By definition, the specific heat provides information on the variation of the internal energy in the system due to a change of temperature. As a consequence, the temperature dependence of the specific heat and the anomaly can be then explained by a specific structure of the excitation energy spectrum of the system. In particular, the anomaly often reveals the presence of unpopulated states in the spectrum of very different kinds of systems. This is the case of the two-level model with the energy gap $\Delta$ in which the specific heat experiences a peak whose value is of the order of $N k_B$, where $N$ is the number of atoms, and which is located at the anomaly temperature $k_B T_A \approx 0.4\Delta$. This effect is known as *Schottky Anomaly* [7]. Similar phenomena can be observed in solid-state and lattice systems as long as the thermal energy is comparable with the gap and for which a small temperature increase induces a significant change in the specific heat. At higher temperatures, the levels are instead evenly populated resulting in lower specific heat. Schottky-like anomalies emerge in systems of very different nature: Bose-Hubbard

model [8], metals [9], crystals [10], compounds [11], quantum ferrimagnets [12] and in spin systems [13–15] even simulating the anomaly in black holes [16]. An external magnetic field changes $\Delta$ and then $T_A$, and the shape of the peak in the specific heat, as observed [17].

Another class of systems that exhibit an anomaly is restricted to one spatial dimensionality. We focus here on the paradigmatic one-dimensional (1D) Bose gas with contact repulsive interactions and which exhibits a complicated spectrum [18]. At low momenta, the linear phononic dispersion determines the low-temperature thermodynamics. An excellent description for this low-energy regime is provided by the Luttinger Liquid theory which is valid for any interaction strength [19]. As the temperature increases, higher momenta get explored and the deviation of the spectrum from the phononic behavior becomes relevant [20, 21], resulting in a continuous structure delimited by two branches of elementary excitations. The upper Lieb I and lower Lieb II branches are associated with the particle and hole excitations, respectively. In particular, Lieb II branch does not have any counterpart for bosonic ensembles in higher spatial dimensions. The existence of a peak in the temperature dependence of the specific heat in a 1D Bose gas is known from the thermal Bethe-Ansatz solution [22] which provides an exact numerical result for this integrable system although it does not give any physical explanation for the anomaly. The anomaly cannot be either interpreted in terms of a thermal phase transition which does not occur because of the 1D geometry [1]. In addition, the complicated structure of the spectrum has not permitted so far an easy interpretation of its effects on the corresponding behavior of thermodynamic quantities like the specific heat. We will show that the presence of the anomaly is related to an important change in the structure of the excitations which is exactly described by the dynamic structure factor (DSF). Previous studies on the DSF of a 1D Bose gas have relied on the Bethe-Ansatz method [23] and interpreted the excitations in terms of particles and holes [24]. In the strongly-repulsive regime, the DSF has been obtained by using the Bose-Fermi mapping [25, 26] and by performing a perturbative expansion on the inverse interaction strength [27]. However, most of the results were limited at temperatures lower than the anomaly value $T_A$ and do not provide then any insight into the peak of the specific heat.

In this work, we report the presence of a peak in the temperature-dependence of the specific heat at constant volume for any finite interaction strength of a 1D Bose gas. We solve numerically the thermal Bethe-Ansatz equations within the Yang-Yang theory, which give an exact result of the specific heat at all temperatures $T$ and interaction strengths [22]. Then, we investigate analytically different tractable regimes, with the use of several perturbative theories holding at low and high temperature, and weak and strong interactions. We demonstrate that the peak can be interpreted in terms of a novel anomaly effect, never reported so far in ultracold gases, and which provides a fundamental insight into the importance of the intrinsic features of the complicated spectrum. This work contains both a qualitative physical interpretation for the anomaly in terms of the presence of unpopulated states in the spectrum which simulate an energy gap as well as a quantitative explanation based on the behavior of the DSF in a wide range of temperatures. Most importantly, both descriptions hold for any value of the interaction strength. We show that the new kind of anomaly shares the thermodynamic features of the Schottky analogue as in both cases the value of the specific heat is of the order of $N k_B$ and the anomaly temperature $T_A$ can be expressed as

$$k_B T_A(\gamma) \sim \Delta(\gamma) \,, \tag{1}$$

as being proportional and of the same order of the "energy gap" $\Delta$. We have made explicit the dependence on the interaction strength $\gamma$ characterizing the new anomaly and which plays the role of the magnetic field in the Schottky analogue. The origins of the two anomalies are related to the spectral properties and, in particular, to the presence of unpopulated states. While in solid-state systems $\Delta$ is provided by the energy gap between the two lowest levels in the

spectrum, in a 1D Bose gas we interpret $\Delta$ as the energy of the maximum of the Lieb II hole-like branch below which unpopulated states are present at zero temperature. We refer to this phenomenon then as *Hole Anomaly*. The hole energy $\Delta$ behaves as an energy gap at the thermodynamic level, by inducing the anomaly in the temperature-dependence of the specific heat. In order to quantify the structure of the excitations, we calculate the DSF for a wide range of temperatures and interaction strengths using the ab-initio Path Integral Monte Carlo numerical method. Our results show that the anomaly temperature $T_A$ determines a critical threshold at which a significant change in the structure of the DSF occurs. While for temperature below $T_A$ the description of excitations in terms of quasiparticles holds, at higher temperatures it fails, as it occurs in any superfluid phase transition around the critical temperature, which is indeed forbidden in 1D systems. The breakdown of the quasiparticle description is due to the thermal broadening around $T_A$ of the peak of the DSF as a function of frequency and for a fixed wavenumber. These results for the DSF are general as they are observed to be valid for several values of the interaction strength.

## 2  Model

A 1D gas of $N$ bosons with contact repulsive interactions is described by the Hamiltonian

$$H = -\frac{\hbar^2}{2m} \sum_{i=1}^{N} \frac{\partial^2}{\partial x_i^2} + g \sum_{i>j}^{N} \delta(x_i - x_j), \tag{2}$$

where $m$ is the atom mass, $g = -2\hbar^2/(ma) > 0$ is the 1D coupling constant, and $a < 0$ is the 1D $s$-wave scattering length. The dimensionless interaction strength $\gamma = -2/(na)$ depends on the gas parameter $na$, with $n = N/L$ the linear density and $L$ the length of the system. There is a continuous interaction crossover which encompasses different quantum degeneracy regimes. In the Gross-Pitaevskii (GP) regime of weak repulsion $\gamma \ll 1$ and of high density $n|a| \gg 1$ the gas admits a mean-field description [28]. In the Tonks-Girardeau (TG) regime of very strong repulsion $\gamma \gg 1$ and low density $n|a| \ll 1$, bosons become impenetrable and the system wavefunction can be mapped onto that of an ideal Fermi gas (IFG), resulting in identical thermodynamic behavior [29].

The Lieb-Liniger model describes the system at zero temperature, where the ground-state energy $E_0$, chemical potential $\mu_0 = (\partial E_0/\partial N)_{a,L}$ and sound velocity $v = \sqrt{n/m(\partial \mu_0/\partial n)_a}$ can be exactly calculated from the Bethe-Ansatz method as a function of $\gamma$ [18, 28, 30]. The sound velocity smoothly changes from the mean-field $v_{GP} = \sqrt{gn/m}$ to the Fermi value $v_F = \hbar\pi n/m$ in the TG regime.

## 3  Specific Heat

Within the canonical ensemble, the thermodynamics of a 1D Bose gas is captured by the Helmholtz free energy $A = E - TS$, with $E$ the internal energy and $S = -(\partial A/\partial T)_{a,N,L}$ the entropy. The chemical potential is defined as $\mu = (\partial A/\partial N)_{T,a,L}$ and the specific heat at constant volume (or length $L$, in 1D) is

$$C = (\partial E/\partial T)_{a,N,L}. \tag{3}$$

Figure 1 shows with symbols the specific heat per particle as a function of the temperature $T$ and for characteristic values of the interaction strength $\gamma$, obtained from the thermal Bethe-Ansatz (TBA) equations. In this figure, the temperature $\tau = T/T_F$ is rescaled by the Fermi value $T_F$ provided by the corresponding energy $E_F = k_B T_F = \hbar^2 \pi^2 n^2/(2m)$. The peak in the specific

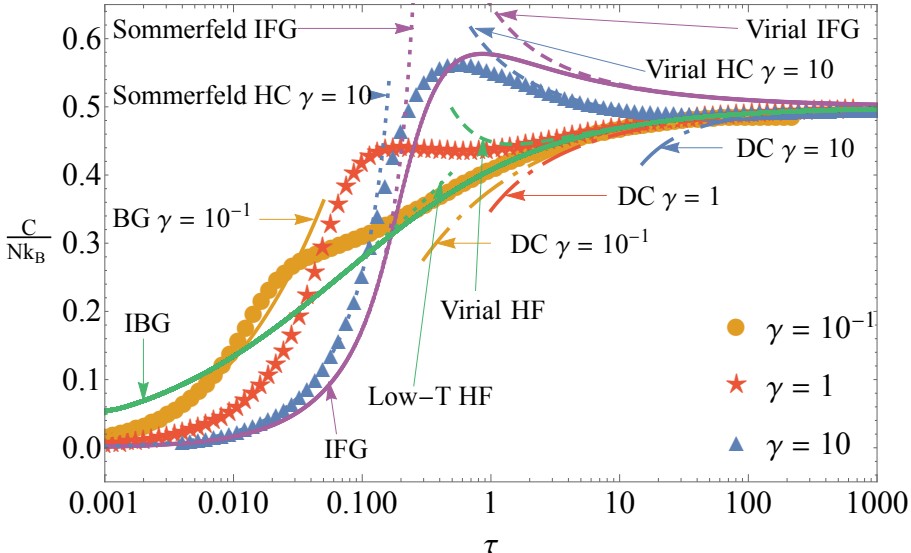

Figure 1: Specific heat at constant length per particle vs temperature in Fermi units $\tau = T/T_F$ and in the thermodynamic limit. The symbols denote numerical thermal Bethe-Ansatz results for several interaction strengths $\gamma$. Lines correspond to the analytical theories. The ideal Bose gas (IBG), Hartree-Fock (HF) and ideal Fermi gas (IFG) descriptions are independent on the value of $\gamma$.

heat is more defined and located at higher temperature $T_A$ by approaching the fermionized TG regime of large $\gamma$. We provide below the understanding of dominant effects in the regimes of the specific heat which may be treated analytically and which show an excellent agreement with TBA in Fig. 1. The following limits are particularly important for the interpretation of Fig. 1 as a diagram in terms of $\gamma$ and $T$ of the many different regimes.

For weak interactions $\gamma \ll 1$, a reliable description is provided by Hartree-Fock (HF) theory, which yields the chemical potential $\mu_{\text{HF}} = \mu_{\text{IBG}} + 2gn$ [28], where $\mu_{\text{IBG}}$ is the corresponding value of the ideal Bose gas (IBG). The thermodynamic quantities depend then on the coupling constant $g$ only through the contribution at zero temperature, and the specific heat, Eq. (3), is the same as that of the IBG. The low-$T$ expansion of the equation of state in terms of the effective fugacity close to unity $\tilde{z} = e^{\beta(\mu_{\text{HF}} - 2gn)} \approx 1$ [31] gives (see Appendix B.1 for the complete derivation)

$$\frac{C_{\text{HF}}}{Nk_B} \approx \frac{3}{8} \frac{\zeta(3/2)}{\zeta(1/2)} M(\tau) \left[ 1 - \frac{\pi M(\tau) D(\tau)}{\zeta(1/2)\zeta(3/2)} \right], \tag{4}$$

where $M = \sqrt{\pi\tau}\zeta(1/2)$, $\zeta(x)$ is the Riemann zeta function and $D = \left(3M^2 - 18M + 32\right) \times (M-2)^{-3}$. Equation (4) agrees with TBA for $T \lesssim T_d$ where $T_d = T_F/\pi^2$ is the quantum degeneracy temperature. At high temperatures $T \gg T_d$, the virial expansion for $\tilde{z} \ll 1$ provides (see Appendix B.2)

$$\frac{C_{\text{HF}}}{Nk_B} = \frac{1}{2} \left[ 1 - \frac{n\lambda}{4\sqrt{2}} - \frac{2\sqrt{3}-5}{16\sqrt{2}} (n\lambda)^3 + O(n\lambda)^5 \right], \tag{5}$$

where $\lambda = \sqrt{2\pi\hbar^2/(mk_BT)}$ is the thermal wavelength. Eq. (5) is an expansion for $n\lambda \ll 1$.

In the weakly-interacting regime at low temperatures $T/T_d \ll \sqrt{\gamma} \ll 1$ [32], the gas behaves as a quasicondensate and its thermodynamics can be described by the Bogoliubov

(BG) theory in terms of a gas of non-interacting bosonic quasi-particles [28, 33]. The specific heat per particle is (see Appendix C)

$$\frac{C_{\text{BG}}}{Nk_B} = \frac{1}{n(k_B T)^2} \int_{-\infty}^{+\infty} \frac{dp}{2\pi\hbar} \epsilon^2(p) \frac{e^{\beta\epsilon(p)}}{\left[e^{\beta\epsilon(p)} - 1\right]^2}, \tag{6}$$

where $\beta = (k_B T)^{-1}$ and $\epsilon(p) = \sqrt{p^2 v^2 + [p^2/(2m)]^2}$ is the BG spectrum at zero temperature [18, 30], which depends on $\gamma$ through the sound velocity $v$. Within the Luttinger Liquid (LL) theory of very low temperatures, one considers only the phononic contribution to the BG dispersion, $\epsilon(p) \approx v|p|$, and obtains the universal result $C_{\text{LL}}/(Nk_B) = \pi k_B T/(3n\hbar v)$ [34] which is valid for the whole interaction crossover [19]. Our finding for the LL regime corrects a misprint in Ref. [28]. The next-to-leading term in the low-$p$ expansion of the BG spectrum, $\epsilon(p) \approx v|p|\left[1 + p^2/(8m^2 v^2)\right]$, allows to calculate the first correction beyond Luttinger Liquid [33] $C_{\text{BG}} = C_{\text{LL}}[1 - 3\pi^2(k_B T)^2/(10m^2 v^4) + O(T^4)]$ (see Appendix C.1).

For $T/T_d \gg \max\left(1, \gamma^2\right)$, one enters into the decoherent classical (DC) regime where both phase and density fluctuations are large and the gas approaches the IBG behavior at high temperature [32]. The specific heat per particle is (see Appendix D)

$$\frac{C_{\text{DC}}}{Nk_B} \approx \frac{1}{2} - \left(1 + \frac{1}{2\sqrt{2}}\right)\frac{n\lambda}{4\pi} - \frac{3\gamma^2}{32\pi\sqrt{2}}(n\lambda)^3. \tag{7}$$

For strong interactions $\gamma^2 \gtrsim \max\left(\pi^2, T/T_d\right)$, the thermodynamics can be understood via the hard-core (HC) model [33, 35]. The specific heat is then obtained from that of an ideal Fermi gas, subtracting from the system size $L$ a "negative excluded volume" $Na$, where the diameter of the HC is equal to the scattering length $a < 0$:
$C_{\text{HC}}(L) = C_{\text{IFG}}(L \to \hat{L} \equiv L - Na)$. Following the Sommerfeld expansion of the IFG specific heat [36] holding for $\tau \ll 1$ and reported in Appendix E.1, we get the low-temperature expansion

$$\frac{C_{\text{HC}}}{Nk_B} = \frac{\pi^2}{6}\hat{\tau}\left[1 + \frac{2}{5}\pi^2\hat{\tau}^2 + \frac{35}{36}\pi^4\hat{\tau}^4 + O\left(\hat{\tau}^6\right)\right], \tag{8}$$

where $\hat{\tau} = k_B T/\hat{E}_F$. The effective Fermi energy $\hat{E}_F = \hbar^2\pi^2\hat{n}^2/(2m)$ depends on the rescaled density $\hat{n} = n/(1 - an)$ which considers the HC correction and it is valid for $n|a| \ll 1$. At high temperature $\pi^2 < T/T_d \lesssim \gamma^2$, we apply the virial expansion to an IFG (see Appendix E.2), and we derive the high-$T$ behavior of the specific heat per particle:

$$\frac{C_{\text{HC}}}{Nk_B} = \frac{1}{2}\left[1 + \frac{\hat{n}\lambda}{4\sqrt{2}} - \frac{2\sqrt{3} - 5}{16\sqrt{2}}(\hat{n}\lambda)^3 + O(\hat{n}\lambda)^4\right] + B, \tag{9}$$

where the shift $B = -\left(2\sqrt{2} + 1\right)/\left[4\sqrt{2\pi}(\gamma + 2)\right]$ corresponds to the $O(n\lambda)$-term of the DC regime, Eq. (7), calculated at the upper temperature bound of the HC approximation $T = \gamma^2 T_d$ and with the density replaced by its rescaled value $n \to \hat{n}$. The shift ensures the continuous crossover between the virial HC and the DC regimes [32]. Differently from the virial expansions, Eqs. (5) and (9), where the thermal wavelength is much larger than the absolute value of the scattering length $\lambda \gg |a|$, in the DC regime, Eq. (7), $\lambda < |a|$. The TG regime ($\gamma = +\infty$) which has the same thermodynamic properties of an IFG, is then recovered from the HC model when $a = 0$, but it is not connected with the DC regime at high $T$. In fact, by approaching the TG limit, while the validity range of temperature of the virial HC theory gets broader $T/T_d < +\infty$ and $B \to 0$, the DC regime disappears as its condition $T/T_d \gg \gamma^2$ is not longer satisfied, see Fig. 1.

# 4 Hole Anomaly

At a microscopic level, the underlying mechanism responsible for the appearance of the anomaly in the specific heat is represented with a sketch in Fig. 2. It portrays the main features of the dynamic structure factor of a 1D Bose gas for temperature smaller and close to the anomaly threshold $T_A$. At zero temperature, the DSF exhibits a continuous structure which is delimited by the Lieb I and II branches [37] corresponding to the particle and hole excitation dispersion, respectively, for any interaction strength $\gamma$. In the spectrum, there are then states which are not populated and some of them are located below the lower Lieb II hole branch. The thermal fluctuations smear the borders of the DSF by an amount of energy equal to $k_B T$. At low temperature $T < T_A$, the quasiparticle picture of the excitations is valid. At small momenta, quasiparticles are provided by phonons with linear dispersion $\omega(k) = v|k|$ where $\omega$ is the frequency and $k$ is the wavenumber. At high temperature $T > T_A$, it is no longer possible to identify a single dominant excitation and the quasiparticle picture breaks down. The temperature $T_A$ where this occurs can be estimated by the value of the "gap" $\Delta$ at the inflection point of the hole branch located at the Fermi wavenumber $k_F = \pi n$, so that $k_B T_A \sim \Delta$, see also Eq. (1). The quantity $\Delta$ corresponds then to the energy of the maximum of the lower Lieb II branch. Up to the same level of accuracy, the value of $\Delta$ can be approximated by the typical energy of phonons calculated at the inflection point $\Delta \propto v\hbar k_F$ [28, 38], so that we obtain

$$k_B T_A \sim v\hbar k_F \,, \tag{10}$$

which has been derived from a microscopic description.

The spectrum at zero temperature can be calculated from the exact Bethe-Ansatz method [18]. For any value of the interaction strength $\gamma$, the relevant hole excitation for the anomaly is always located at the maximum of the Lieb II branch at $k_F$, while its energy value $\Delta$ changes in the interaction crossover. We report in Fig. 3 the comparison between the hole energy $\Delta$ and the temperature $T_A$ of the peak in the specific heat, to test the validity of the proposed microscopic anomaly mechanism. The hole energy $\Delta$ has been calculated with Bethe

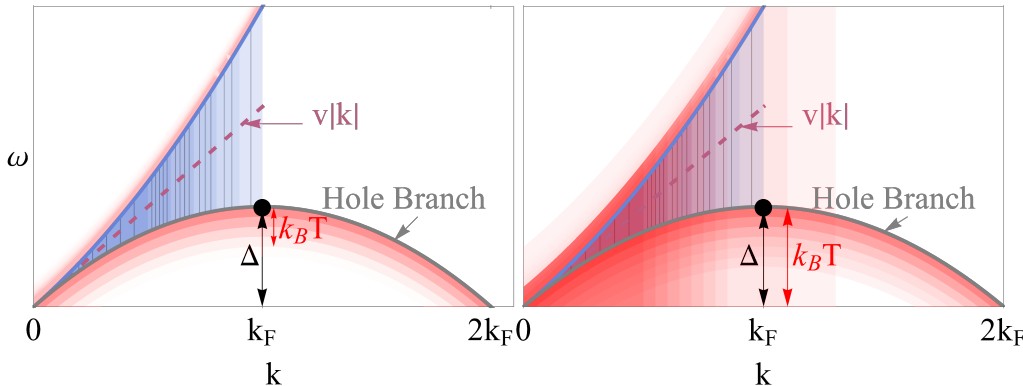

Figure 2: Sketch of the dynamic structure factor at a temperature below (left) and around (right) the value of the Hole Anomaly. Upper particle-like Lieb I and lower hole-like Lieb II branches are reported with solid curves. Dashed line denotes the linear phononic spectrum $\omega(k) = v|k|$ where $\omega$ is the frequency, $v$ is the sound velocity and $k$ is the wavenumber. The hole excitation responsible for the anomaly is located at the Fermi wavenumber $k_F$ and its energy is equal to $\Delta$. The dynamic structure factor at zero temperature is reported with the blue shaded region with vertical lines. Its thermal contribution is instead denoted with red shading.

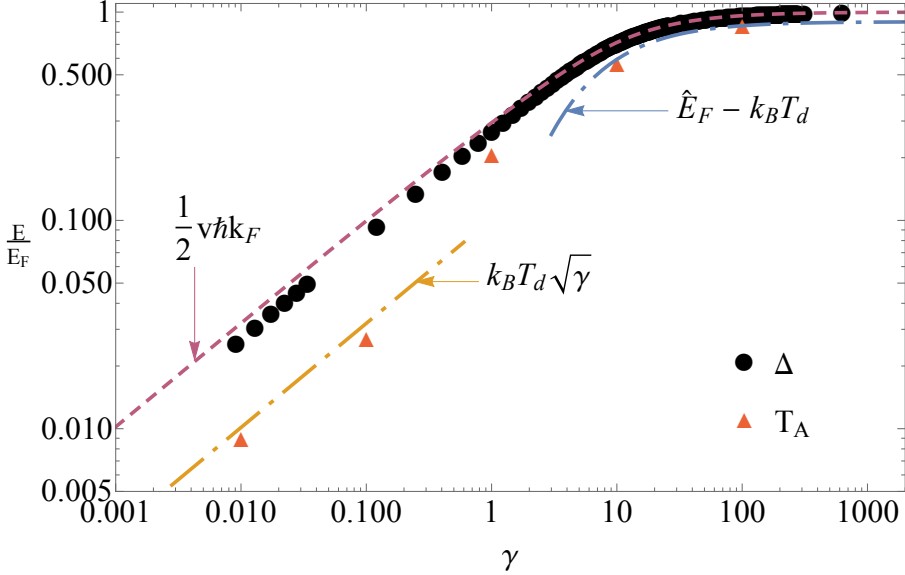

Figure 3: Hole energy $\Delta$ and anomaly temperature $T_A$ in Fermi units vs interaction strength $\gamma$. Circles denote numerical Bethe-Ansatz results for $\Delta$. Triangles represent $T_A$ estimated from the anomalies in the specific heat shown in Fig. 1. Dot-dashed lines correspond to the upper-temperature bounds of the BG and Sommerfeld HC theories, for small and large values of $\gamma$ respectively. Dashed line shows the phononic energy scale at the Fermi wavenumber $k_F$.

Ansatz [39, 40]. We also show the energy scales at which the anomaly occurs for small and large values of $\gamma$. Such scales correspond to the upper-temperature bounds of the validity of the BG, Eq. (6), and Sommerfeld HC, Eq. (8), theories, the latter of which has been shifted by the degeneracy energy $k_B T_d$. The temperature ranges of the validity of the BG and Sommerfeld HC approaches are $T/T_d \ll \sqrt{\gamma}$ and $k_B T \ll \hat{E}_F$, respectively. We finally report half of the phononic energy calculated at the Fermi wavenumber $v\hbar k_F/2$. This provides a universal energy scale for the hole excitation $\Delta \approx v\hbar k_F/2$ for any $\gamma$ [38]. Overall, a good agreement between $\Delta$ and $T_A$ is found (keeping in mind that their relation is through a coefficient of the order of unity that depends on $\gamma$) even if $\gamma$ is changed by more than four orders of magnitude. This fully supports the hole scenario for the anomaly, described by Eq. (1).

This important result solves the open problem of relating the features of the microscopic complicated excitation spectrum with the thermodynamic behavior of a 1D Bose gas. In fact, similarly to the Schottky anomaly in the two-level model, the new hole anomaly emerges from the presence of unpopulated states in the spectrum. The close analogy to the Schottky anomaly is further reinforced if one associates the energy $\Delta$ of the maximum of the hole branch with the energy gap in the two-level system. Indeed, both phenomena share a similar proportionality relation for the anomaly temperature $k_B T_A \sim \Delta$, see also Eq. (1), where $T_A$ and $\Delta$ are of the same order in both cases. While in the Schottky anomaly all parameters appearing in the above proportionality relation can be changed by applying an external magnetic field, in a 1D Bose gas they all depend on the interaction strength $\gamma$ controlled by the magnetic Fano-Feshbach resonances in experiments [20]. By approaching the strongly-correlated Tonks-Girardeau regime with very large interaction strength $\gamma$, $k_B T_A \approx \Delta$, Fig. 3. The reduction of the discrepancy between the anomaly temperature $T_A$ and the hole energy $\Delta$ for strong interactions is explained with the presence of just one physical energy set by the Fermi value $E_F = k_B T_F = \hbar^2 \pi^2 n^2/(2m)$ in the fermionized Tonks-Girardeau regime and which provides

the common limit for $T_A$ and $\Delta$ for large $\gamma$. Finally, the monotonic increasing behavior of the "energy gap" $\Delta$ with $\gamma$ in Fig. 3 determines the anomaly peak in the specific heat getting more defined and located at higher temperatures $T_A$ by increasing the interaction strength, see Fig. 1.

## 4.1 Chemical Potential

We argue here that the anomaly is a universal property of all 1D atomic gases, even if they are described by a different Hamiltonian from the one we have considered, Eq. (2). A thermal feature seen as an abrupt change in the dependence with temperature might be found not only in the specific heat but also in other interesting thermodynamic quantities like the chemical potential, see Fig. 4. Indeed, at temperatures much larger than the interaction energy, one can

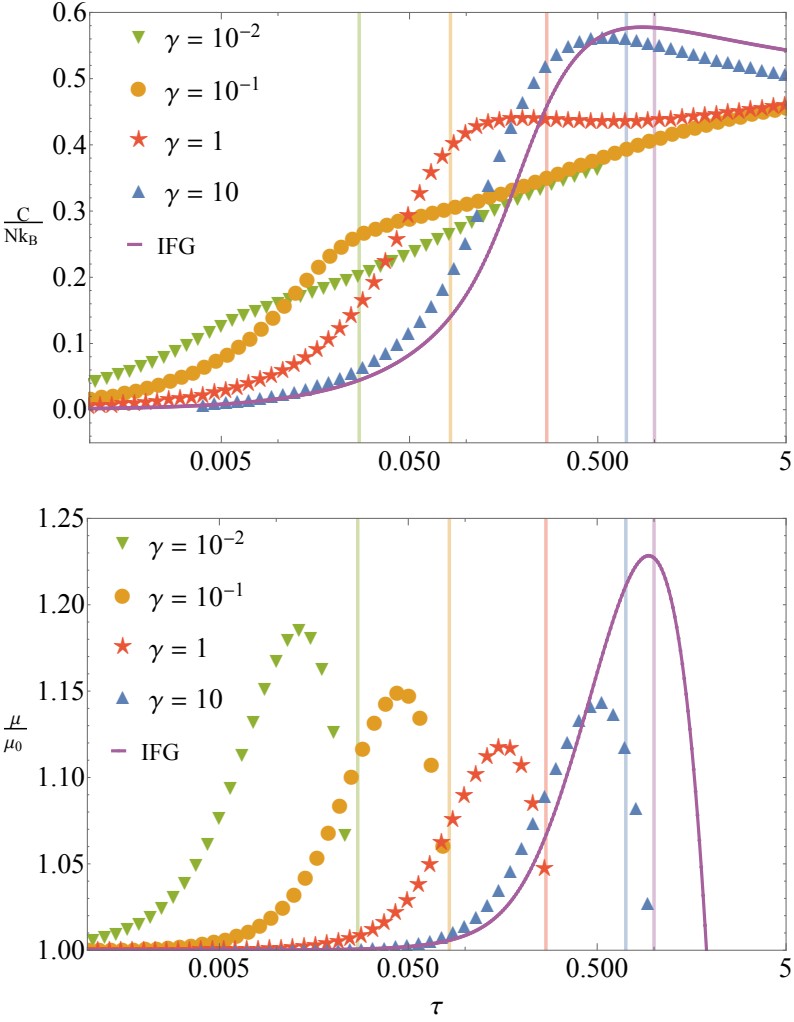

Figure 4: Specific heat at constant length per particle (upper panel) and chemical potential rescaled by its value at zero temperature $\mu_0$ (lower panel) vs temperature in Fermi units $\tau = T/T_F$. The symbols denote numerical thermal Bethe-Ansatz results for several interaction strengths $\gamma$. Solid line corresponds to the ideal Fermi gas (IFG) finding. Vertical lines represent the hole energy $\Delta$ and they are reported in an increasing order of $\gamma$ from low (left) to high (right) values.

approximately consider the gas as being ideal. The chemical potential is negative and it is a decreasing function for both ideal Bose and Fermi gases in the limit of high temperature. As it was demonstrated by some of the authors of the present paper in Ref. [19], for a wide class of 1D systems with a gapless spectrum, the presence of the phononic excitations whose dispersion is linear $v|p|$ leads to a $T^2$-increase in the chemical potential in the Luttinger Liquid regime of low temperatures. As the chemical potential is an increasing and a decreasing function at low and high temperatures, respectively, it must exhibit a maximum. Within the Luttinger Liquid formalism, the Fermi energy $E_F = k_B T_F$ is a relevant scale for both fermionic and bosonic gases with the same density $n$ and atomic mass $m$. The characteristic anomaly temperature at which the maximum of the chemical potential appears can be estimated by setting the thermal $T^2$-correction equal to the dominant contribution $\mu_0/E_F \approx (T/T_F)^2$, provided by the chemical potential at zero temperature and which can be expressed in terms of the sound velocity $\mu_0 \propto mv^2$ for any value of the interaction strength. By combining the above approximate conditions one exactly recovers Eq. (10), representing the characteristic temperature of the anomaly in the specific heat. In fact, even if the values of the anomaly temperature $T_A$ are different in the chemical potential and in the specific heat, they are of the same order and both can be then well approximated by Eq. (10) which has been obtained here from thermodynamic considerations rather than a microscopic description.

For sake of clarity, Fig. 4 shows with symbols the specific heat per particle and the chemical potential as a function of the temperature $T$ and for characteristic values of the interaction strength $\gamma$, obtained from the thermal Bethe-Ansatz equations. Solid line represents the ideal Fermi gas result. Vertical lines denote the hole energy $\Delta$ for different values of $\gamma$. We notice that both the specific heat and the chemical potential exhibit an anomaly for any value of the interaction strength $\gamma$. In addition, for both these thermodynamic quantities, the discrepancy between the anomaly temperature $T_A$, corresponding to the position of the thermal feature, and the hole energy $\Delta$ decreases by approaching the fermionized TG regime of large $\gamma$, as depicted also in Fig. 3. The chemical potential as a function of temperature has been previously calculated with thermal Bethe-Ansatz method in the strongly-interacting regime [26] and for different values of the interaction strength [19, 33].

## 5 Dynamic Structure Factor

Quantitative information on the excitation spectrum of the system is provided by the dynamic structure factor $S(k, \omega)$ which describes the dynamic response with frequency $\omega$ of a quantum many-body system to a weak density perturbation with wavenumber $k$. It is defined by the Fourier transform of the real-time $t$ density-density correlation function

$$S(k, \omega) = \frac{1}{N} \int_{-\infty}^{+\infty} \frac{dt}{2\pi} e^{i\omega t} \text{Tr}[n_T \rho_{-k}(t)\rho_k(0)],$$ (11)

where $\rho_k(t) = e^{iHt/\hbar} \rho_k e^{-iHt/\hbar}$ is the time evolution, following the Hamiltonian $H$ of Eq. (2), of the density perturbation operator $\rho_k = \sum_{i=1}^{N} e^{-ikx_i}$. The thermal density matrix $n_T = e^{-\beta H}/Z$, where $Z = \text{Tr}(e^{-\beta H})$ is the partition function, allows for the calculation of the expectation value of any quantum observable $O$: $\langle O \rangle = \text{Tr}(n_T O)$.

We have calculated the DSF for different temperatures $T$ and interaction strengths $\gamma$ by employing the Path Integral Monte Carlo (PIMC) technique, see Sec. 6 and Appendix G. We consider here the results for the intermediate regime $\gamma = 1$, which cannot be handled with perturbative theories in Fig. 1. However, the behavior of the DSF described below is the same for different values of $\gamma$, see Appendix G.

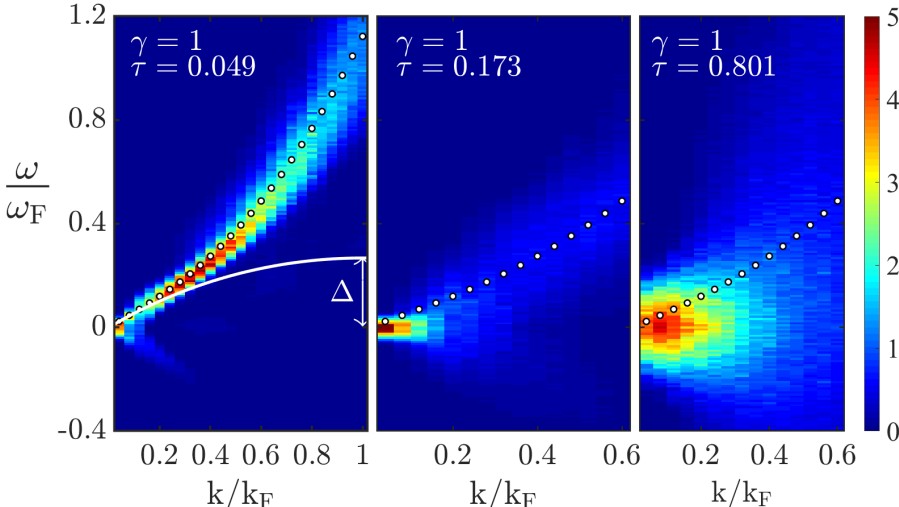

Figure 5: Dynamic structure factor for the interaction strength $\gamma = 1$. Solid line shows the energy of the hole branch at zero temperature calculated with exact Bethe Ansatz and its value for $k = k_F$ gives its maximum energy $\Delta$. Path Integral Monte Carlo numerical results are for: i) the dynamic structure factor which is represented with the heatmap in units of the inverse of the Fermi frequency $\omega_F = E_F/\hbar$ and for different temperatures $\tau = T/T_F$; ii) the single-mode frequency $\omega_{SM}$ is denoted by dots whose sizes are larger than the error bars. Wavenumber $k$ is in units of the Fermi value $k_F$.

In Fig. 5 we provide characteristic examples of the DSF at temperatures below, around and above the anomaly value $T_A$. For $T < T_A$, the DSF resembles its behavior at zero temperature [23]. At small wavenumber $k$, the thermal excitations stay in the Luttinger Liquid regime and the DSF exhibits a sharp peak located at a frequency $\omega$ which satisfies the linear phononic law $\omega(k) = v|k|$. As $k$ increases, the DSF structure gets broader, but it remains centered at a $\omega > 0$ value, while the $\omega < 0$ contributions are yet negligible. For all $k$ values, the peak of the DSF is then well approximated by the Feynman relation, for which the full excitation spectrum is described in terms of a coherent single-mode (SM) quasi-particle [41]:

$$\hbar\omega_{SM}(k) = \hbar^2 k^2 / [2mS(k)] \ , \tag{12}$$

where $S(k) = \int_{-\infty}^{+\infty} d\omega S(k,\omega)$ is the static structure factor. In Fig. 5, we show for comparison the single-mode frequency $\omega_{SM}$ obtained within the Feynman approximation based on PIMC results for $S(k)$. As the temperature $T$ is increased, the zero-temperature approximation no longer provides a reliable description for the excitations of the gas. At $T \approx T_A$, the DSF does not show anymore the phononic behavior at small $k$, but it exhibits instead a signal around $\omega = 0$. At large $k$, there is a broad distribution centered at $\omega_{SM}$. For $T > T_A$, the DSF shows a very broad signal with the maximum centered at $\omega = 0$ for any value of $k$. At temperatures above the anomaly, the dynamics of the system is not then described by a coherent single excitation with a finite frequency.

To quantify the DSF in the temperature crossover, in Fig. 6 we report $S(k,\omega)$ as a function of frequency $\omega$ and temperature $T$ and for fixed values of the wavenumber $k$. This crossover is explained in terms of the interplay between quantum correlations (i.e. interaction effects) and thermal fluctuations in the dynamics of the excitations of the system. At temperatures below that of the anomaly $T < T_A$, quantum correlations are more important than thermal fluctuations which are then treated as a small perturbation. Above the anomaly threshold

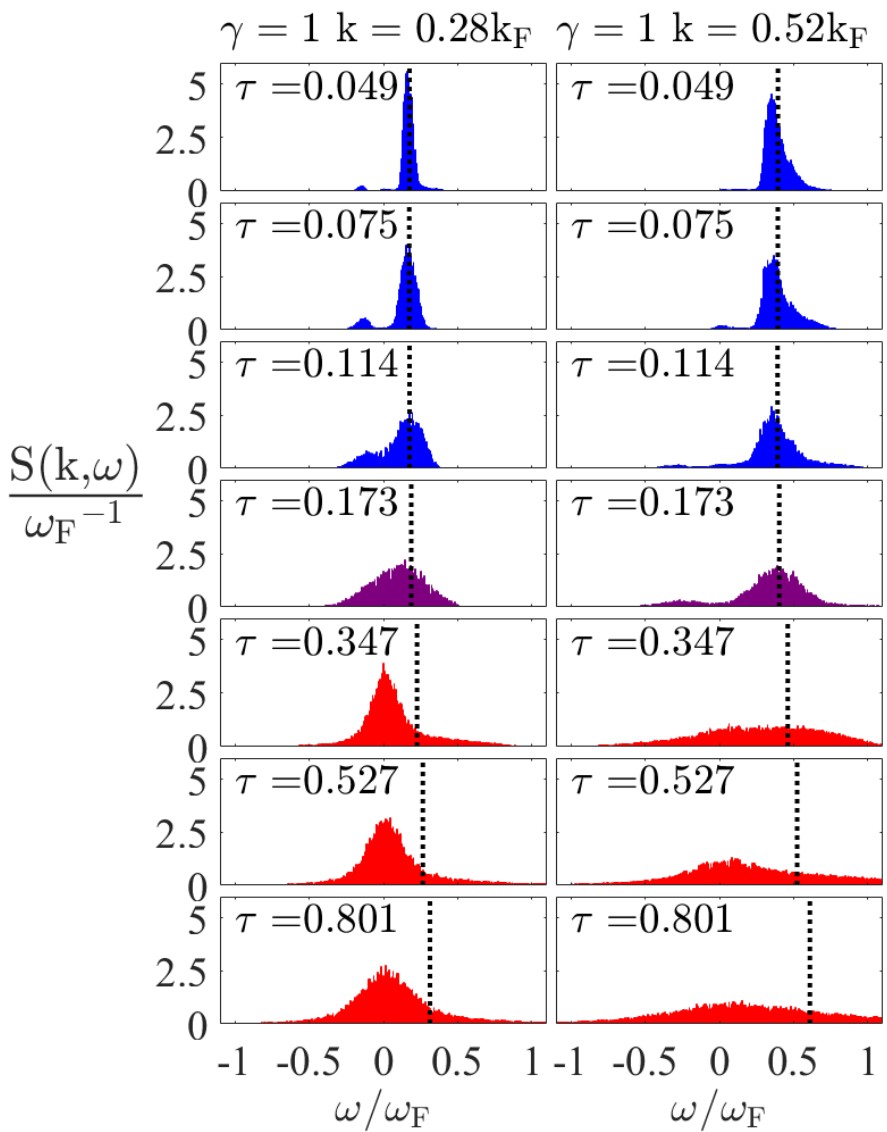

Figure 6: Path Integral Monte Carlo results of the dynamic structure factor vs frequency $\omega$ for two values of the wavenumber $k$ (left and right column) and interaction strength $\gamma = 1$. Each row corresponds to different temperatures $\tau = T/T_F$. Vertical line denotes the single-mode frequency $\omega_{SM}$. Colors represent the regimes below ($\tau \le 0.114$), around ($\tau \approx 0.173$) and above ($\tau \ge 0.347$) the anomaly.

$T > T_A$, the DSF is characterized by a broad incoherent component and thermal fluctuations dominate over quantum effects. In the non-trivial intermediate regime around the anomaly $T \approx T_A$, where simple analytical theories cannot be applied, our results show the coexistence of quantum correlations and thermal fluctuations whose contributions are comparable.

As can been seen from Figs. 5-6, the frequency dependence of the DSF for fixed values of the wavenumber $k$ at $T < T_A$ is characterized by a sharp peak, located at frequency $\omega_{SM}$ which signals the excitation of a single mode making the quasiparticle description, Eq. (12), valid for any value of $k$. At high temperatures $T > T_A$, many different modes are excited and the resulting DSF exhibits a broad structure as a function of frequency. The breakdown of the quasiparticle picture around the anomaly temperature $T_A$ is then due to the thermal broadening of the peak of the DSF as a function of frequency and for fixed values of the

wavenumber.

It is worth noticing that the superfluid-to-normal phase transition in Bose systems, in two and three spatial dimensions, shows the same trend in the DSF when crossing the critical temperature $T_c$: the quasi-particle collective excitation turns to a broad thermal response for $T > T_c$. Therefore the present 1D case, showing also a finite peak in the specific heat, resembles the critical behavior of Bose systems at higher spatial dimensions. However, differently to two- and three- dimensional geometry in which a true phase transition occurs making the change of the behavior in DSF more evident, in 1D there is instead a temperature crossover which makes the broadening of the peak in the DSF smoother. As a result, the anomaly temperature $T_A$ found from the specific heat dependence, provides only an appropriate energy scale rather than a precise value at which the DSF broadening is observed.

## 6  Path Integral Monte Carlo Method

The PIMC method relies on the description of an ensemble of $N$ quantum atoms in terms of a set of $N$ classical polymers, each of them reproducing the quantum delocalization of one particle [3]. In this way, the thermal average $\langle O \rangle$ of the quantum observable $O$ is expressed as a multidimensional integral which can be efficiently computed with a Monte Carlo algorithm. The PIMC makes use of the convolution property of the propagator which admits an analytical approximation only for small imaginary times. We calculate such approximation by choosing a pair-product scheme which is based on the exact solution of the two-body problem for the Hamiltonian in Eq. (2) [42,43]. In order to recover the Bose statistics of the indistinguishable quantum particles, we sample stochastically the permutations among the atoms with the worm algorithm [44]. PIMC results have been carefully checked by benchmarking the expectation values for the energy per particle and the isothermal compressibility against the exact TBA calculation in Appendix F.

The PIMC method is exact for the calculation of the static properties like the energy [3], but it only allows for an indirect estimation of the dynamic properties, such as the DSF $S(k, \omega)$, Eq. (11). In order to compute $S(k, \omega)$, one has to recover the correlation function in the imaginary time $\varepsilon$: $F(k, \varepsilon) = \mathrm{Tr}[n_T \rho_{-k}(-i\varepsilon)\rho_k(0)]/N$ which is related to the DSF via a Laplace transform $F(k, \varepsilon) = \int_{-\infty}^{+\infty} d\omega S(k, \omega)\left[e^{-\hbar\omega\varepsilon} + e^{-\hbar\omega(\beta-\varepsilon)}\right]$. The inversion of the Laplace transform is a mathematically ill-conditioned problem and make unfeasible a precise reconstruction of the DSF from PIMC data of the $F(k, \varepsilon)$, which are unavoidably affected by statistical uncertainties. Yet, several numerical approaches have been presented in literature to tackle this problem [45–51]. These methods are able to recover the main features of the DSF, like the frequencies and the spectral weight of the main peaks, and have provided insightful results in the study of several quantum many-body systems [52–56]. We reconstruct the DSF by employing a simulated annealing procedure which minimizes the $\chi^2$-deviation between the expectation value of $F(k, \varepsilon)$, obtained from a guess of $S(k, \omega)$, and the PIMC data [51,57]. Details on the annealing method can be found in Appendix H.

## 7  Conclusions

In this work, we provide a detailed description of the new hole anomaly in 1D Bose gases evidenced by a thermal feature in thermodynamic quantities such as the chemical potential and specific heat as a function of temperature. We argue that the presence of the anomaly is due to the region of unpopulated states located below the lower hole branch in the excitation spectrum at zero temperature and which behaves as an energy gap from the thermodynamic

point of view. We show that the anomaly temperature $T_A$ and the energy $\Delta$ of the maximum of the hole branch are functions of the interaction strength $\gamma$ and they are proportional and of the same order: $k_B T_A(\gamma) \sim \Delta(\gamma)$. Another excellent energy scale for the hole anomaly is provided by phonons at the Fermi momentum, which ensures the high control over both $T_A$ and $\Delta$ by finely tuning $\gamma$ through the sound velocity. We provide an additional simple characterization of the anomaly in terms of a qualitative change in the structure of the excitations, governed by a single quasiparticle mode at low temperatures which gets suppressed by thermal fluctuations at higher temperatures. The breakdown of the quasiparticle picture is due to the thermal broadening around the anomaly temperature of the structure of the excitations as a function of frequency. Our description for the hole anomaly is valid for any value of the interaction strength $\gamma$ and solves the open problem of relating the effects of the complicated spectrum to the thermodynamic behavior of a 1D Bose gas. The anomaly is a reminiscence of a phase transition, that is not allowed in 1D systems, and signals a change of quantum regime.

The exact results of the specific heat as a function of temperature and interaction strength $\gamma$ were obtained with the Bethe-Ansatz method and the main features were described analytically. Beyond the validity of different analytical limits, novel quantum regimes emerge. We computed the dynamic structure factor with the ab-initio Path Integral Monte Carlo technique. We carefully characterized then the behavior of the dynamic response with temperature, by showing that quantum correlation and thermal fluctuation contributions are comparable in the anomaly regime for any $\gamma$. The Path Integral Monte Carlo method has been applied to a 1D Bose gas with contact interactions for the first time in our work. Our calculations extend to a wide range of temperatures compared to previous studies which were restricted only to very low temperature.

We show that an anomaly is always present in the chemical potential of any 1D atomic gas of both bosonic and fermionic nature. A similar proportionality relation $k_B T_A(\gamma) \sim \Delta(\gamma)$ applies to the Schottky anomaly in the two-level model where the dependence on the interaction strength $\gamma$ is replaced by the magnetic field and $\Delta$ has to be interpreted as the energy gap. The two-level model is a low-temperature approximation of any discrete spectrum which is found in many different solid-state systems where the dependence of the Schottky anomaly on the magnetic field has been indeed observed.

A hole-like anomaly, whose position and structure change with an external magnetic field, has been also detected in the temperature-dependence of the specific heat in 1D spin chains [58,59]. The continuous excitation spectrum of this system, which exhibits a twofold particle-hole nature similar to the case of the 1D Bose gas, has been experimentally measured showing an excellent agreement with the exact calculation based on the Bethe-Ansatz [60,61]. A similar anomaly may be also found in 1D electronic systems with particle-hole spectrum [62].

In both 1D quantum spin chains and ladders in the presence of a changing magnetic field, the anomaly has been theoretically and experimentally studied not only in the specific heat but also through the minimum in the magnetization as a function of temperature [63–65]. The lack of the $\lambda$-shaped divergence in the specific heat and the analytic minimum in the magnetization reflects the absence of a true phase transition. The temperatures of the two thermal features, although similar, are not identical [64] as can be found by comparing the values of $T_A$ in the specific heat and chemical potential in the 1D Bose gas, see Sec. 4.1. The hole anomaly shows important analogies with anomalies present in these spin systems in the limit of spinless fermions corresponding to the TG regime in our system. In fact, $T_A$ has been estimated from the spin energy gap of the spectrum, Eq. (1), and it corresponds to the crossover from the Luttinger Liquid to the high-temperature regime due to the competition of the chemical potential at zero temperature and its lowest thermal contribution, see Sec. 4.1. Finally, the anomaly signals the excitation of the states at the bottom of the band in the spectrum, similarly to the corresponding mechanism in the 1D Bose gas depicted in Fig. 2.

Since our description of the hole anomaly is universally valid for any finite interaction strength, it may be applied to model the behavior of anomalies in 1D spin chains and ladders even in the regime of interacting fermions. The dynamic structure factor in spin ladders at zero temperature has been calculated by using the density-matrix renormalization group (DMRG) in real time [65] and an extension at finite temperature is highly desirable [66–68]. Our PIMC results can then give a qualitative insight into the dynamical correlations around the anomaly temperature in spin systems. Differently to the DMRG technique which can be employed only in lattice systems, PIMC method can be applied in both discretized and continuous limits, the latter of which is the case of the present work.

A tantalizing possibility is that the hole anomaly could be employed as a quantum simulator as it provides an in-depth understanding of diverse anomalies in other more complicated many-body systems such as atoms, solids, electrons, spin chains and ladders. The new anomaly shares typical properties of a quantum simulator as its future observation is feasible and it can be achieved in clean experimental atomic settings where precise control and broad tunability of the interaction strength $\gamma$ are possible. In addition, the applicability of several exact methods to the 1D Bose gas provides fundamental insights into the problem. This concept of quantum simulation [69] at the thermodynamic level aims at the understanding of strongly-correlated systems, the development of innovative materials, and the emergence of new quantum technologies.

At the experimental level, the temperature-dependence of the specific heat has been observed in a three-dimensional strongly-interacting Fermi gas, where the detection of the peak allowed for a precise measurement of the critical temperature of the superfluid phase transition [5]. The chemical potential as a function of temperature and interaction strength $\gamma$ has been measured in a 1D Bose gas, resulting in an excellent agreement with thermal Bethe-Ansatz solution [70]. The optical tube trap allowed the exploration of the chemical potential at temperatures both below and above the anomaly, by keeping satisfied the condition for the 1D geometry [70]. The employed experimental technique in both measurements is the in-situ absorption imaging whose signal-to-noise ratio has been recently enhanced [71]. In 1D, three-body losses are strongly reduced [72], and the spatial uniform density can be achieved with a flatbox potential [73]. All these premises make the observation of the novel hole anomaly in a 1D Bose gas particularly appealing for current experimental settings [70]. At fixed interaction strength $\gamma$, the detection of the anomaly may be employed as a precise in-situ temperature probe. The $\gamma$-dependence of the anomaly can be observed by applying Fano-Feshbach resonances [20, 74]. The dynamic structure factor of a 1D Bose gas has been probed using Bragg spectroscopy [20, 21], but at temperatures smaller than the anomaly threshold.

Looking forward, in harmonically trapped 1D Bose gases, the specific heat determines the time-dependence of the temperature in hydrodynamic breathing modes [75]. Hole anomaly may play then a key role in the hydrodynamic-collisionless transition which has been predicted with the evolution of breathing modes due to an increase of temperature [76]. This intriguing idea may explain the mismatch between the predictions and the measurements of the breathing mode frequencies through the crossover between the regimes of the quasicondensate and the ideal Bose gas [77]. The HC description, which is valid close to the strongly-correlated TG regime where the anomaly effect is enhanced, is expected to hold even in very different systems at low density. The anomaly should be well visible then even for positive scattering length $a > 0$ in: i) short-range interacting systems like the metastable super Tonks-Girardeau gas [78, 79], ii) finite-range interacting ensembles of dipolar [80, 81] and Rydberg atoms [82], 1D bosonic $^4$He (liquid) [56] and 1D fermionic $^3$He (gas) [83]. Our findings can also be extended to 1D liquids in bosonic mixtures [84, 85] at finite temperature [86].

# Acknowledgements

The authors gratefully acknowledge G. Lang and J.-S. Caux for the insightful suggestions. They also thank in particular the unknown Referee 3 for the many important comments especially on the analogies of their findings with the anomalies in spin chains and ladders and the strengths of the PIMC method if compared with the DMRG technique.

**Author contributions**    G. D. R., G. E. A. and J. B. devised the initial concepts and theory. Analytical results were derived by G. D. R. Monte Carlo numerical simulations were implemented by R. R. Bethe-Ansatz exact calculations were performed by G. E. A. The manuscript was written by G. D. R. with suggestions from G. E. A., R. R. and J. B. The project was supervised by J. B. and G. E. A.

**Funding information**    G. D. R.'s received funding from the European Union's Horizon 2020 research and innovation program under the Marie Skłodowska-Curie grant agreement *UltraLiquid* No. 797684 and with the grant IJC2020-043542-I funded by MCIN/AEI/10.13039/501100011033 and by "European Union NextGenerationEU/PRTR". G. D. R., G. E. A. and J. B. were partially supported by grant PID2020-113565GB-C21 funded by MCIN/AEI/10.13039/501100011033, the Spanish MINECO (FIS2017-84114-C2-1-P), and the Secretaria d'Universitats i Recerca del Departament d'Empresa i Coneixement de la Generalitat de Catalunya within the ERDF Operational Program of Catalunya (project QuantumCat, Ref. 001-P-001644).

**Data and materials availability**    Data corresponding to the ideal Bose and Fermi gas in Fig. 1 may be found available online at https://upcommons.upc.edu/handle/2117/353128. All the other data needed to evaluate the conclusions are present in the main text and/or the Appendices. Additional data related to this work may be requested to the authors (giulia.de.rosi@upc.edu).

# A    Main Quantities

From the Helmholtz free energy $A = E - TS$, where $E$ is the internal energy and $T$ the temperature, we calculate the entropy

$$S = -(\partial A/\partial T)_{a,N,L}\,, \tag{13}$$

and the pressure

$$P = -(\partial A/\partial L)_{T,a,N} = n(\mu - A/N)\,, \tag{14}$$

where $\mu$ is the chemical potential. The Tan's contact parameter [33] is defined in systems with zero-range interactions

$$\mathcal{C} = \frac{4m}{\hbar^2}\left(\frac{\partial A}{\partial a}\right)_{T,N,L} = \frac{4nN}{a^2}g_2(0)\,, \tag{15}$$

and it is proportional to the normalized pair correlation function [28] at zero relative distances $x = 0$

$$g_2(x = x_1 - x_2) = \frac{\langle \hat{\psi}^+(x_2)\,\hat{\psi}^+(x_1)\,\hat{\psi}(x_1)\,\hat{\psi}(x_2)\rangle}{n^2}\,, \tag{16}$$

where $\hat{\psi}(x)$ is the field operator.

# B Hartree-Fock theory for a weakly-interacting Bose gas

In this Appendix, we provide details about the low- and the high-temperature expansions within the Hartree-Fock approximation.

The equation of state for a 1D weakly-interacting Bose gas with density $n$ and pressure $P$ can be obtained from

$$\begin{cases} n\lambda = g_{1/2}(\tilde{z}), \\ P\lambda = gn^2\lambda + k_B T g_{3/2}(\tilde{z}), \end{cases} \tag{17}$$

where $\tilde{z} = e^{\beta(\mu_{\mathrm{HF}}-2gn)}$ is the effective fugacity within the Hartree-Fock (HF) theory [28], $\lambda = \sqrt{2\pi\hbar^2/(mk_B T)}$ is the thermal wavelength and the Bose functions are

$$g_\nu(z) = \frac{1}{\Gamma(\nu)} \int_0^{+\infty} dx \frac{x^{\nu-1}}{z^{-1}e^x - 1}, \tag{18}$$

where $\Gamma(\nu)$ is the Euler gamma function.

## B.1 Low-temperature expansion

The Bose functions can be approximated for $\tilde{z} = e^{-\alpha} \approx 1$, with small and positive $\alpha$ [31]

$$g_\nu(e^{-\alpha}) = \Gamma(1-\nu)\alpha^{\nu-1} + \sum_{i=0}^{+\infty} (-1)^i \frac{\zeta(\nu-i)}{i!}\alpha^i, \tag{19}$$

where $\zeta(x)$ is the Riemann zeta function.

By inverting the expression for $n$ in Eq. (17) where we employ Eq. (19), we obtain the following approximation for the effective fugacity

$$\tilde{z} \approx e^{-\pi\left[\frac{\sqrt{\pi\tau}}{\sqrt{\pi\tau}\zeta(1/2)-2}\right]^2}, \tag{20}$$

where $\tau = k_B T/E_F$, and $E_F = k_B T_F = \hbar^2\pi^2 n^2/(2m)$ is the Fermi energy. By making use of the definition of $\tilde{z}$ in Eq. (20) and by considering only the real solution, we find the low-temperature behavior of the chemical potential:

$$\mu_{\mathrm{HF}} \approx 2gn - E_F\left[\frac{\pi\tau}{\sqrt{\pi\tau}\zeta(1/2)-2}\right]^2. \tag{21}$$

By combining Eqs. (19) - (20) in the equation of the pressure $P$, given by Eq. (17), we obtain

$$P_{\mathrm{HF}} \approx gn^2 + \frac{\sqrt{\pi}}{2}nE_F\zeta(3/2)\tau^{3/2}\left\{1 - \frac{\pi^{3/2}\sqrt{\tau}}{\zeta(3/2)}\frac{\left[3\sqrt{\pi\tau}\zeta(1/2)-4\right]}{\left[\sqrt{\pi\tau}\zeta(1/2)-2\right]^2}\right\}. \tag{22}$$

From Eqs. (14) and (21)-(22), we calculate the low-temperature expansion of the free energy per particle

$$\frac{A_{\mathrm{HF}}}{N} \approx gn - \frac{\sqrt{\pi}}{2}E_F\zeta(3/2)\tau^{3/2}\left\{1 - \frac{3\pi^{3/2}\sqrt{\tau}}{\zeta(3/2)\left[\sqrt{\pi\tau}\zeta(1/2)-2\right]}\right\}, \tag{23}$$

from which we get the corresponding entropy per particle, Eq. (13)

$$\frac{S_{\mathrm{HF}}}{Nk_B} \approx \frac{3}{4}\zeta(3/2)\sqrt{\pi\tau}\left\{1 - \frac{\pi^{3/2}\sqrt{\tau}}{\zeta(3/2)}\frac{\left[3\sqrt{\pi\tau}\zeta(1/2)-8\right]}{\left[\sqrt{\pi\tau}\zeta(1/2)-2\right]^2}\right\}, \tag{24}$$

and the energy per particle

$$\frac{E_{\text{HF}}}{N} \approx gn + \frac{\sqrt{\pi}}{4} E_F \zeta(3/2) \tau^{3/2} \left\{ 1 - \frac{3\pi^{3/2}\sqrt{\tau}}{\zeta(3/2)} \frac{\left[\sqrt{\pi\tau}\zeta(1/2)-4\right]}{\left[\sqrt{\pi\tau}\zeta(1/2)-2\right]^2} \right\}. \tag{25}$$

We notice that in the HF approximation, all the thermodynamic quantities depend on the interactions only through their contribution at zero temperature. The Tan's contact per particle, Eq. (15), is then independent on temperature:

$$\frac{\mathcal{C}_{\text{HF}}}{N} = 2n^3\gamma^2, \tag{26}$$

and we obtain the pair correlation function of an ideal Bose gas $g_2(0)_{\text{HF}} = 2$.

## B.2 High-temperature virial expansion

The Bose functions, Eq. (18), admit the series representation in terms of small effective fugacity $\tilde{z} \ll 1$: $g_\nu(\tilde{z}) = \sum_{i=1}^{+\infty} \tilde{z}^i/i^\nu$. By inverting the expression for $n$ in Eq. (17) and by expanding for $n\lambda \ll 1$, we get

$$\tilde{z} = n\lambda - \frac{(n\lambda)^2}{\sqrt{2}} + \frac{\sqrt{3}-1}{\sqrt{3}}(n\lambda)^3 + O[(n\lambda)^4]. \tag{27}$$

By making use of the definition of $\tilde{z}$ in Eq. (27) and an expansion for $n\lambda \ll 1$, we obtain the virial expansion of the chemical potential:

$$\mu_{\text{HF}} = k_B T \left[ \ln(n\lambda) - \frac{n\lambda}{\sqrt{2}} + \frac{3\sqrt{3}-4}{4\sqrt{3}}(n\lambda)^2 - \frac{2\sqrt{3}-5}{6\sqrt{2}}(n\lambda)^3 + O[(n\lambda)^4] \right] + 2gn. \tag{28}$$

From the equation of $P$, Eq. (17), we derive the high-temperature behavior of the pressure:

$$P_{\text{HF}} = nk_B T \left[ 1 - \frac{n\lambda}{2\sqrt{2}} + \frac{3\sqrt{3}-4}{6\sqrt{3}}(n\lambda)^2 - \frac{5}{4}\frac{(2\sqrt{3}-5)}{6\sqrt{2}}(n\lambda)^3 + O[(n\lambda)^4] \right] + gn^2. \tag{29}$$

The expansion of the free energy per particle is

$$\frac{A_{\text{HF}}}{N} = k_B T \left[ \ln(n\lambda) - 1 - \frac{n\lambda}{2\sqrt{2}} + \frac{3\sqrt{3}-4}{4\sqrt{3}}\frac{(n\lambda)^2}{3} + \frac{2\sqrt{3}-5}{6\sqrt{2}}\frac{(n\lambda)^3}{4} + O[(n\lambda)^4] \right] + gn, \tag{30}$$

which has been previously derived at the order $O[(n\lambda)^2]$ [33]. The entropy and the energy per particle are, respectively

$$\frac{S_{\text{HF}}}{Nk_B} = -\left[ \ln(n\lambda) - \frac{3}{2} - \frac{n\lambda}{4\sqrt{2}} - \frac{2\sqrt{3}-5}{6\sqrt{2}}\frac{(n\lambda)^3}{8} + O[(n\lambda)^5] \right], \tag{31}$$

$$\frac{E_{\text{HF}}}{N} = \frac{1}{2}k_B T \left[ 1 - \frac{n\lambda}{2\sqrt{2}} + \frac{3\sqrt{3}-4}{6\sqrt{3}}(n\lambda)^2 + \frac{2\sqrt{3}-5}{8\sqrt{2}}(n\lambda)^3 + O[(n\lambda)^4] \right] + gn. \tag{32}$$

# C Bogoliubov theory for a weakly-interacting Bose gas at low temperature

We report here the calculation of the low-temperature expansion of the Bogoliubov theory.

The thermodynamics of a quasicondensate can be described in terms of a gas of noninteracting bosonic quasiparticles [28, 33], by applying the Bogoliubov (BG) theory, whose free energy per particle is

$$\frac{A_{\text{BG}}}{N} = \frac{E_0}{N} + \frac{k_B T}{n} \int_{-\infty}^{+\infty} \frac{dp}{2\pi\hbar} \ln\left[1 - e^{-\beta\epsilon(p)}\right], \tag{33}$$

where $E_0$ is the ground-state energy obtained within the Lieb-Liniger theory at zero temperature [30], $\epsilon(p) = \sqrt{p^2 v^2 + [p^2/(2m)]^2}$ is the $T = 0$ BG spectrum [18, 30] and $\beta = (k_B T)^{-1}$. From Eq. (33), we can calculate the complete thermodynamics of the system, such as the entropy per particle, Eq. (13)

$$\frac{S_{\text{BG}}}{Nk_B} = \frac{1}{n} \int_{-\infty}^{+\infty} \frac{dp}{2\pi\hbar} \left[\frac{\beta\epsilon(p)}{e^{\beta\epsilon(p)} - 1} - \ln\left(1 - e^{-\beta\epsilon(p)}\right)\right], \tag{34}$$

and the energy per particle

$$\frac{E_{\text{BG}}}{N} = \frac{E_0}{N} + \frac{1}{n} \int_{-\infty}^{+\infty} \frac{dp}{2\pi\hbar} \frac{\epsilon(p)}{e^{\beta\epsilon(p)} - 1}. \tag{35}$$

The chemical potential, the pressure and the Tan's contact parameter have been derived in Ref. [33].

## C.1 Low-temperature expansion from non-linear Bogoliubov spectrum

The low-momentum expansion of the Bogoliubov dispersion relation
$x = \epsilon(p) = v|p|\left[1 + p^2/(8m^2v^2)\right] > 0$ can be inverted to get the real and positive solution $p$. Hence, for the free energy per particle, Eq. (33), we obtain the integral:

$$\int_0^{+\infty} dx \frac{1}{v\left[1 + \frac{3p^2(x)}{8(mv)^2}\right]} \ln\left(1 - e^{-\frac{x}{k_B T}}\right) \approx -\frac{\pi^2}{6} \frac{(k_B T)}{v} + \frac{\pi^4}{120} \frac{(k_B T)^3}{m^2 v^5}, \tag{36}$$

where the analytic result may be found expanding the integrand for $|p| \ll mv$, which is valid at low temperatures. We get the low-$T$ behavior of the free energy per particle, within the Bogoliubov theory [33]:

$$\frac{A_{\text{BG}}}{N} = \frac{E_0}{N} - \frac{\pi}{6} \frac{(k_B T)^2}{\hbar n v} \left[1 - \frac{\pi^2}{20} \frac{(k_B T)^2}{m^2 v^4} + O(T^4)\right], \tag{37}$$

from which we calculate the entropy and the energy per particle

$$\frac{S_{\text{BG}}}{Nk_B} = \frac{\pi}{3} \frac{k_B T}{\hbar n v} \left[1 - \frac{\pi^2}{10} \frac{(k_B T)^2}{m^2 v^4} + O\left(T^4\right)\right], \tag{38}$$

$$\frac{E_{\text{BG}}}{N} = \frac{E_0}{N} + \frac{\pi}{6} \frac{(k_B T)^2}{\hbar n v} \left[1 - \frac{3\pi^2}{20} \frac{(k_B T)^2}{m^2 v^4} + O\left(T^4\right)\right]. \tag{39}$$

The Tan's contact per particle, Eq. (15), is [33]

$$\frac{\mathcal{C}_{\text{BG}}}{N} = \frac{\mathcal{C}_0}{N} + \frac{\bar{\mathcal{C}}}{N} \frac{(k_B T)^2}{m^2 v^4} \left[1 - \frac{\pi^2}{4} \frac{(k_B T)^2}{m^2 v^4} + O\left(T^4\right)\right], \tag{40}$$

where the factor $\bar{\mathcal{C}} = \pi m^3 v^2 N \gamma^2 (\partial v/\partial\gamma)_n/(3\hbar^3)$ depends on the sound velocity $v = \sqrt{gn/m}$ and the value at zero temperature $\mathcal{C}_0/N = n^3\gamma^2$ has to be compared with that of the HF theory,

Eq. (26). The pair correlation function approaches the unity corresponding to the value of the coherent regime:

$$g_2(0)_{\mathrm{BG}} = 1 + \frac{\pi^5}{24} \frac{\tau^2}{\gamma^{3/2}} \left[ 1 - \frac{\pi^6}{16} \frac{\tau^2}{\gamma^2} + O\left(\tau^4\right) \right], \tag{41}$$

where $\tau = T/T_F$. Our finding, Eq. (41), agrees with the result at the mean-field level of Ref. [32], and we derived the additional $O\left(T^4\right)$ correction.

## D  Decoherent Classical regime

We derive in the following the thermodynamic quantities in the decoherent classical regime.

The pair correlation function in the decoherent classical (DC) regime is close to the HF value $g_2(0)_{\mathrm{HF}} = 2$ [32]

$$g_2(0)_{\mathrm{DC}} \approx g_2(0)_{\mathrm{HF}} - \frac{\gamma}{\sqrt{2}} n\lambda, \tag{42}$$

from which we calculate the Tan's contact per particle, Eq. (15)

$$\frac{\mathcal{C}_{\mathrm{DC}}}{N} \approx \frac{\mathcal{C}_{\mathrm{HF}}}{N} \left( 1 - \frac{\gamma}{2\sqrt{2}} n\lambda \right), \tag{43}$$

where we have used Eq. (26). By integrating the above equation with $\lambda < |a|$ and $a < 0$, we find the free energy per particle

$$\frac{A_{\mathrm{DC}}}{N} = k_B T \left[ \ln(n\lambda) - 1 \right] + \frac{1}{N} \int_a^\lambda da \frac{\partial A_{\mathrm{DC}}}{\partial a}$$

$$\approx k_B T \left[ \ln(n\lambda) - 1 \right] - \frac{\hbar^2 n^2 \gamma}{m} \left[ 1 + \left( 2 + \frac{1}{\sqrt{2}} \right) \frac{1}{\gamma n\lambda} - \frac{\gamma}{4\sqrt{2}} n\lambda \right], \tag{44}$$

where the constant of the integration has been chosen equal to the leading classical gas value whose prefactor is provided by the thermal energy $k_B T$. The entropy, Eq. (13), and the energy per particle are, respectively

$$\frac{S_{\mathrm{DC}}}{N k_B} \approx - \left[ \ln(n\lambda) - \frac{3}{2} \right] + \left( 1 + \frac{1}{2\sqrt{2}} \right) \frac{n\lambda}{2\pi} + \frac{\gamma^2}{16\pi\sqrt{2}} (n\lambda)^3, \tag{45}$$

$$\frac{E_{\mathrm{DC}}}{N} \approx \frac{1}{2} k_B T \left[ 1 - \left( 1 + \frac{1}{2\sqrt{2}} \right) \frac{n\lambda}{\pi} \right] - \frac{\hbar^2 n^2 \gamma}{m} \left( 1 - \frac{3\gamma}{8\sqrt{2}} n\lambda \right). \tag{46}$$

## E  Ideal Fermi gas

In this Appendix, we provide details about the derivation of the Sommerfeld and the virial expansions of an ideal Fermi gas.

The equation of state of a 1D ideal Fermi gas (IFG) with density $n$ and pressure $P$ can be found from

$$\begin{cases} n\lambda = f_{1/2}(z), \\ P\lambda = k_B T f_{3/2}(z), \end{cases} \tag{47}$$

where we have defined the fugacity $z = e^{\mu/(k_B T)}$ and the Fermi functions

$$f_\nu(z) = \frac{1}{\Gamma(\nu)} \int_0^{+\infty} dx \frac{x^{\nu-1}}{z^{-1} e^x + 1}, \tag{48}$$

where $\Gamma(\nu)$ is the Euler Gamma function.

### E.1 Low-temperature Sommerfeld expansion

The Sommerfeld expansion [36] allows for the calculation of integrals of the form:

$$\int_0^{+\infty} d\epsilon\, H(\epsilon) f(\epsilon) = \int_0^\mu d\epsilon\, H(\epsilon) + \sum_{i=1}^{+\infty} a_i (k_B T)^{2i} \frac{d^{2i-1}}{d\epsilon^{2i-1}} H(\epsilon)|_{\epsilon=\mu}, \tag{49}$$

where

$$f(\epsilon) = \frac{1}{e^{\frac{\epsilon-\mu}{k_B T}} + 1} \tag{50}$$

is the Fermi-Dirac distribution. We consider the 1D density of states of the IFG:

$$H(\epsilon) = \frac{1}{2\sqrt{E_F \epsilon}}, \tag{51}$$

where $E_F = k_B T_F = \hbar^2 \pi^2 n^2 / (2m)$ is the Fermi energy. In Eq. (49), we have introduced

$$a_i = \left(2 - \frac{1}{2^{2(i-1)}}\right)\zeta(2i), \tag{52}$$

where $\zeta(i)$ is the Riemann zeta function.

At very low temperature, the chemical potential of the IFG approaches the Fermi energy and we set then $\mu \to E_F(1+\delta)$ with $0 \le \delta \ll 1$. We take into account the Sommerfeld expansion, Eq. (49), up to the $O(\delta^3)$-order, corresponding to the integer $i = 3$, and we impose the normalization condition $\int_0^{+\infty} d\epsilon\, H(\epsilon) f(\epsilon) = 1$. We solve the resulting equation for the real solution $\delta$ and we expand in series, finding the chemical potential

$$\mu_{\mathrm{IFG}} = E_F \left(1 + \frac{\pi^2}{12}\tau^2 + \frac{\pi^4}{36}\tau^4 + \frac{7\pi^6}{144}\tau^6 + O(\tau^8)\right), \tag{53}$$

with $\tau = k_B T / E_F$.

The Sommerfeld expansion, Eq. (49), enables one to get the low-temperature behavior $k_B T \ll \mu$ of the Fermi functions $f_\nu\left(e^{\frac{\mu}{k_B T}}\right) \approx \frac{1}{\nu\Gamma(\nu)}\left(\frac{\mu}{k_B T}\right)^\nu \left[1 + \frac{\pi^2}{6}\nu(\nu-1)\left(\frac{k_B T}{\mu}\right)^2\right]$. By using the latter expression in Eq. (47), one recovers Eq. (53), and obtains the pressure:

$$P_{\mathrm{IFG}} = \frac{2}{3} n E_F \left[1 + \frac{\pi^2}{4}\tau^2 + \frac{\pi^4}{20}\tau^4 + \frac{35\pi^6}{432}\tau^6 + O(\tau^8)\right]. \tag{54}$$

From Eqs. (53)-(54) and (14), we calculate the low-temperature expansion of the free energy per particle:

$$\frac{A_{\mathrm{IFG}}}{N} = \frac{E_F}{3}\left(1 - \frac{\pi^2}{4}\tau^2 - \frac{\pi^4}{60}\tau^4 - \frac{7\pi^6}{432}\tau^6 + O(\tau^8)\right), \tag{55}$$

the entropy per particle, Eq. (13)

$$\frac{S_{\mathrm{IFG}}}{N k_B} = \frac{\pi^2}{6}\tau \left(1 + \frac{2\pi^2}{15}\tau^2 + \frac{7\pi^4}{36}\tau^4 + O\left(\tau^6\right)\right), \tag{56}$$

the energy per particle

$$\frac{E_{\mathrm{IFG}}}{N} = \frac{E_F}{3}\left(1 + \frac{\pi^2}{4}\tau^2 + \frac{\pi^4}{20}\tau^4 + \frac{35\pi^6}{432}\tau^6 + O\left(\tau^8\right)\right), \tag{57}$$

and the specific heat per particle, Eq. (3), which corresponds to Eq. (8) with zero scattering length $a = 0$.

The Tan's contact per particle, Eq. (15), is [33]:

$$\frac{\mathcal{C}_{\text{IFG}}}{N} = \frac{4m}{\hbar^2} P_{\text{IFG}},\tag{58}$$

where $P_{\text{IFG}}$ is given by Eq. (54). The pair correlation function approaches zero in the IFG limit $\gamma \to \infty$:

$$g_2(0)_{\text{IFG}} = \frac{4}{3}\frac{\pi^2}{\gamma^2}\left[1 + \frac{\pi^2}{4}\tau^2 + \frac{\pi^4}{20}\tau^4 + \frac{35\pi^6}{432}\tau^6 + O(\tau^8)\right],\tag{59}$$

and it agrees with the finding of Ref. [32] at the order $O(\tau^2)$, but we have derived higher thermal corrections.

## E.2 High-temperature virial expansion

The Fermi functions, Eq. (48), can be approximated as $f_\nu(z) = \sum_{i=1}^{+\infty}(-1)^{i-1}z^i/i^\nu$ for small fugacity $z \ll 1$. By inverting the equation for the density $n$, Eq. (47), and by expanding for $n\lambda \ll 1$, we calculate

$$z(n\lambda) = n\lambda + \frac{(n\lambda)^2}{\sqrt{2}} + \frac{\sqrt{3}-1}{\sqrt{3}}(n\lambda)^3 + O[(n\lambda)^4],\tag{60}$$

from which, employing the definition of $z$ and an additional expansion for $n\lambda \ll 1$, we derive the chemical potential:

$$\mu_{\text{IFG}} = k_B T\left[\ln(n\lambda) + \frac{n\lambda}{\sqrt{2}} + \frac{3\sqrt{3}-4}{4\sqrt{3}}(n\lambda)^2 + \frac{2\sqrt{3}-5}{6\sqrt{2}}(n\lambda)^3 + O[(n\lambda)^4]\right].\tag{61}$$

By considering Eq. (60) in the equation of $P$, Eq. (47), we find the high-temperature behavior of the pressure

$$P_{\text{IFG}} = nk_B T\left[1 + \frac{n\lambda}{2\sqrt{2}} + \frac{3\sqrt{3}-4}{6\sqrt{3}}(n\lambda)^2 + \frac{2\sqrt{3}-5}{8\sqrt{2}}(n\lambda)^3 + O[(n\lambda)^4]\right],\tag{62}$$

and the virial expansion of the free energy per particle:

$$\frac{A_{\text{IFG}}}{N} = k_B T\left[\ln(n\lambda) - 1 + \frac{n\lambda}{2\sqrt{2}} + \frac{3\sqrt{3}-4}{6\sqrt{3}}\frac{(n\lambda)^2}{2} + \frac{2\sqrt{3}-5}{6\sqrt{2}}\frac{(n\lambda)^3}{4} + O[(n\lambda)^4]\right].\tag{63}$$

The entropy per particle is:

$$\frac{S_{\text{IFG}}}{Nk_B} = -\left[\ln(n\lambda) - \frac{3}{2} + \frac{n\lambda}{4\sqrt{2}} - \frac{2\sqrt{3}-5}{6\sqrt{2}}\frac{(n\lambda)^3}{8} + O(n\lambda)^5\right],\tag{64}$$

and the energy per particle becomes

$$\frac{E_{\text{IFG}}}{N} = \frac{k_B T}{2}\left[1 + \frac{n\lambda}{2\sqrt{2}} + \frac{3\sqrt{3}-4}{6\sqrt{3}}(n\lambda)^2 + \frac{2\sqrt{3}-5}{8\sqrt{2}}(n\lambda)^3 + O(n\lambda)^4\right],\tag{65}$$

from which we derive the virial expansion of the specific heat which corresponds to Eq. (9) with $a = 0$. The Tan's contact is given by Eq. (58) where the pressure is now provided by Eq. (62). The pair correlation function, Eq. (15), is

$$g_2(0)_{\text{IFG}} = \frac{2\pi^2}{\gamma^2}\frac{P_{\text{IFG}}}{nE_F},\tag{66}$$

whose leading term recovers the result of Ref. [32].

# F   Benchmark of the Path Integral Monte Carlo Results

We present here a benchmark for the expectation values of the internal energy $E$ and the isothermal compressibility of a 1D Bose gas at finite temperature which have been calculated with numerical Path Integral Monte Carlo (PIMC) method. Such benchmark is based on the comparison with exact Thermal Bethe-Ansatz (TBA) findings.

PIMC technique provides exact results for the static properties at finite temperature [3]. For a Bose system, the expectation value of an observable $O$ is expressed as a multidimensional integral which is computed via Monte-Carlo sampling of the coordinates $R$:

$$\langle O \rangle = \text{Tr}(n_T O) = \frac{1}{ZN!} \sum_{\mathcal{P}} \int dR\, G(R, \mathcal{P}R; \beta) O(R). \tag{67}$$

In Eq. (67) we have introduced the thermal density matrix $n_T = e^{-\beta H}/Z$ where $Z = \text{Tr}\left(e^{-\beta H}\right)$ is the partition function, $\beta = (k_B T)^{-1}$ is the inverse temperature and $H$ is the Hamiltonian, Eq. (2). In the last equality, we have considered the coordinate representation $G(R_1, R_2; \beta) = \langle R_2 | e^{-\beta H} | R_1 \rangle$ and $O(R) = \langle R | O | R \rangle$ where $R_i = \{x_{1,i}, x_{2,i} \ldots, x_{N,i}\}$ is a set of the coordinates of the $N$ atoms of the system. $G(R_1, R_2; \beta)$ is the Green function propagator describing the evolution in the imaginary time $\beta$ from the initial $R_1$ to the final $R_2$ configuration. The configuration $\mathcal{P}R$ appearing in Eq. (67) is obtained by applying a permutation $\mathcal{P}$ of the particle labels to the initial configuration $R$ and the sum over the $N!$ permutations allows to take into account the quantum statistics of the identical bosonic atoms.

The key aspect of the Path Integral formalism is the convolution property of the propagator:

$$G(R_1, R_3; \beta_1 + \beta_2) = \int dR_2 G(R_1, R_2; \beta_1) G(R_2, R_3; \beta_2), \tag{68}$$

which can be easily generalized to a series of intermediate steps $R_2 \ldots R_M$ defining a path with $M$ configurations and with total time $\beta = \varepsilon M$, where $\varepsilon$ is the time step. For a finite value of $M$, the path is discretized with time. In the opposite case of very large $M$, the path becomes continuous and $\varepsilon$ approaches zero, corresponding to the limit of high temperatures $T$. In the latter classical limit, the propagator $G$ admits an analytical approximation where the quantum effects of the non-commutativity between the kinetic and interaction potential operators in the Hamiltonian $H$ are neglected. The thermal expectation value, Eq. (67), can be then approximated as

$$\langle O \rangle \simeq \frac{1}{ZN!} \sum_{\mathcal{P}} \int dR_1 \ldots dR_M O(R_1) \prod_{i=1}^{M} G(R_i, R_{i+1}; \varepsilon), \tag{69}$$

where we require the boundary condition $R_{M+1} = \mathcal{P}R_1$. The quantity $p(R_1, \ldots, R_M) = \prod_{i=1}^{M} G(R_i, R_{i+1}; \varepsilon)$ is a probability distribution as it is positive definite and its integral over the space of configurations is equal to the unity. The PIMC approach allows for the evaluation of the integral in Eq. (69) with a stochastic sampling of the $N \times M$ degrees of freedom according to $p(R_1, \ldots, R_M)$. Eq. (69) becomes exact in the limit $M \to \infty$ where the imaginary time $\varepsilon$ is small and the analytical approximation for the propagator $G$ is accurate, by allowing for an exact calculation of the thermal average $\langle O \rangle$ with PIMC method. For this reason, it is important to optimize the number of the convolution terms $M$ in Eq. (69) with a proper benchmark of the PIMC results.

We have calculated the energy per particle $E/N$ with $E = \langle H \rangle$ as a function of temperature and for different values of the interaction strength $\gamma$. The comparison of such PIMC results with the corresponding TBA ones, Fig. 7, provides the estimate of the optimal value for $M$. The excellent PIMC-TBA agreement witnesses the reliability of our numerical findings.

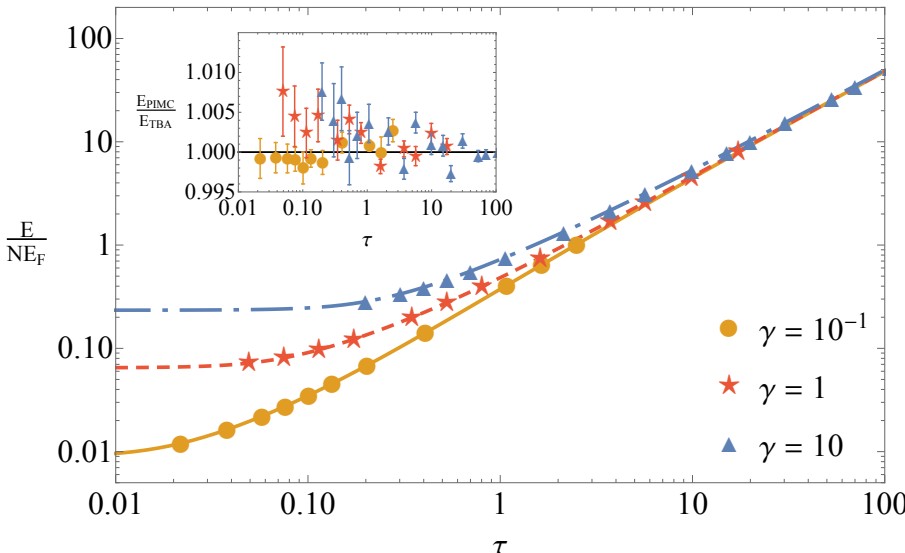

Figure 7: Energy per particle $E/N$ vs temperature for different values of the interaction strength $\gamma$. Symbols denote the Path Integral Monte Carlo (PIMC) results. Lines correspond to the Thermal Bethe-Ansatz (TBA) findings. In the Inset, we report the ratio of the PIMC vs TBA results. Energy and temperature $\tau = T/T_F$ are normalized to the corresponding Fermi values defined by $E_F = k_B T_F$.

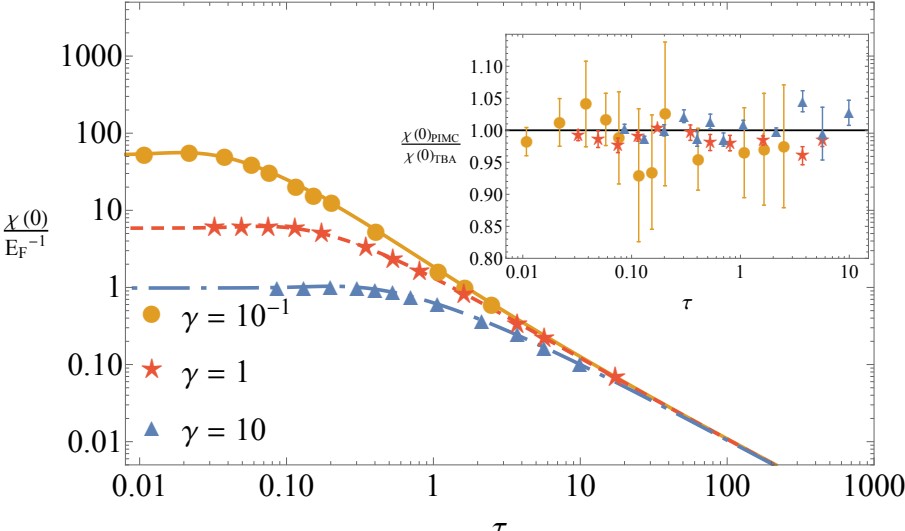

Figure 8: Isothermal compressibility $\chi(0)$ vs temperature for different values of the interaction strength $\gamma$. Symbols denote PIMC results. Lines correspond to the TBA findings. In the Inset, we report the ratio of the PIMC vs TBA results. Isothermal compressibility and temperature $\tau = T/T_F$ are normalized to the corresponding Fermi values $E_F = k_B T_F$.

In order to test the accuracy of the PIMC results for the imaginary-time correlation function $F(k, \varepsilon)$, from which we recover the dynamic structure factor, we have done a similar study for the isothermal compressibility

$$\chi(0) = \left( \frac{\partial n}{\partial P} \right)_{T,a,N} \, , \tag{70}$$

where the pressure $P$ is defined by Eq. (14). The isothermal compressibility corresponds to the zero-wavenumber limit of the static density response function $\chi(k)$ [28], which is related to $F(k, \varepsilon)$:

$$\chi(k) = \int_0^\beta d\varepsilon \, F(k, \varepsilon) \, . \tag{71}$$

We calculate $\chi(k)$ with PIMC procedure from Eq. (71) and we extrapolate $\chi(0)$ from the behavior at smallest $k$. In Fig. 8, we compare the PIMC $\chi(0)$ results with the TBA isothermal compressibility evaluated from Eq. (70) as a function of temperature and for different values of the interaction strength $\gamma$.

## G  Dynamic Structure Factor for different interaction strengths

We report here our PIMC results for the dynamic structure factor (DSF) $S(k, \omega)$ for the interaction strength value $\gamma = 10$. For strong interactions, the DSF distribution gets broader in frequency $\omega$ [23], see Figs. 9-10 which have to be compared with the corresponding ones for $\gamma = 1$ in Figs. 5-6.

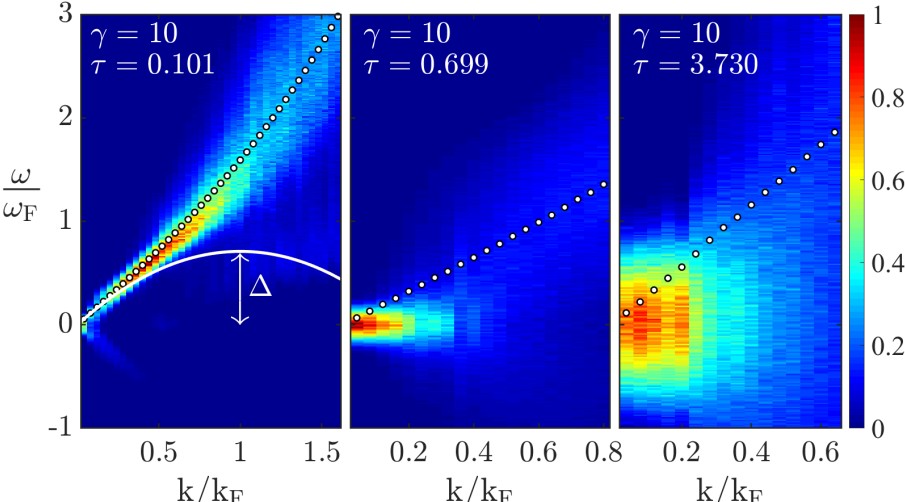

Figure 9: Dynamic structure factor for the interaction strength $\gamma = 10$. Solid line shows the energy of Lieb II branch at zero temperature calculated with exact Bethe-Ansatz and its value for $k = k_F$ gives its maximum energy $\Delta$. Path Integral Monte Carlo numerical results are for: i) the dynamic structure factor which is represented with the heatmap in units of the inverse of the Fermi frequency $\omega_F = E_F/\hbar$ and for different temperatures $\tau = T/T_F$; ii) the single-mode frequency $\omega_{SM}$ is denoted by dots whose sizes are larger than the error bars. Wavenumber $k$ is in units of the Fermi value $k_F$.

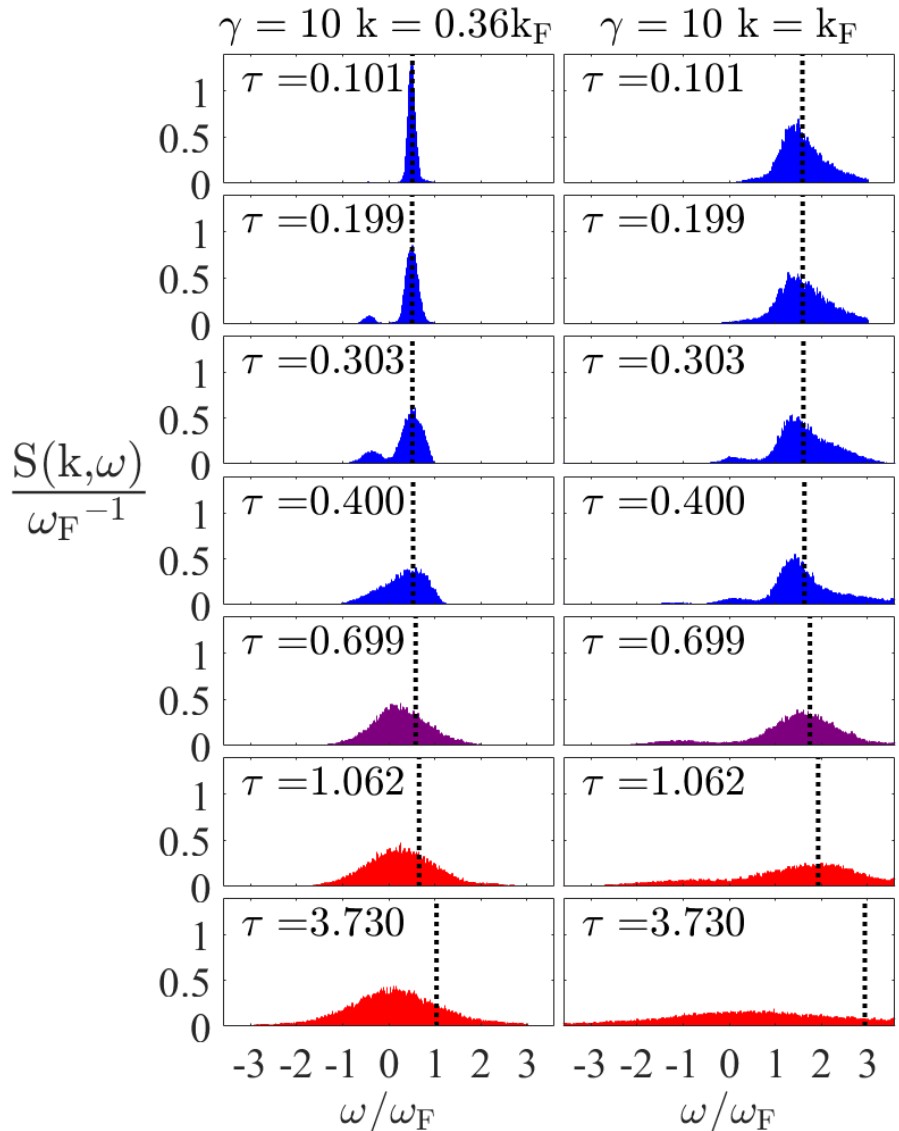

Figure 10: Path Integral Monte Carlo results of the dynamic structure factor vs frequency $\omega$ for two values of the wavenumber $k$ (left and right column) and interaction strength $\gamma = 10$. Each row corresponds to different temperatures $\tau = T/T_F$. Vertical line denotes the single-mode frequency $\omega_{SM}$. Colors represent the regimes below ($\tau \leq 0.4$), around ($\tau \approx 0.699$) and above ($\tau \geq 1.062$) the anomaly.

## H  Simulated Annealing Method

PIMC method simulates the microscopic dynamics of the many-body systems in imaginary-time configuration space. Without access to the real-time evolution, there is no possibility of directly getting the DSF by a Fourier transform of the correlation function or intermediate scattering function $F(k, \varepsilon)$, as it occurs in simulations of classical systems using Molecular Dynamics. Quantum Monte Carlo methods can be conveniently used to sample $F(k, \varepsilon)$ but in imaginary time $\varepsilon$, and from it to get the dynamic response through an inverse Laplace transform. This inverse transform of noisy data is a mathematically ill-posed problem as any error in the input data (statistical, rounding, etc.) is increased exponentially making it impossible

to find a unique solution for the DSF. We have carried out the inverse Laplace transform via the simulated annealing algorithm, which is a well-known stochastic multidimensional optimization method widely used in physics and engineering [57]. This inversion method has been previously applied for the calculation of the DSF in liquid $^4$He across the superfluid-normal phase transition, by showing a reasonable agreement with the experimental data [51].

The dynamic structure factor $S(k, \omega)$ satisfies the detailed balance condition

$$S(k, -\omega) = e^{-\beta \hbar \omega} S(k, \omega),  \tag{72}$$

which relates the dynamic response of negative and positive energy transfers $\hbar \omega$. The correlation function $F(k, \varepsilon)$ is the Laplace transform of the DSF $S(k, \omega)$

$$F(k, \varepsilon) = \int_{-\infty}^{+\infty} d\omega S(k, \omega) \left[ e^{-\hbar \omega \varepsilon} + e^{-\hbar \omega (\beta - \varepsilon)} \right],  \tag{73}$$

where we have used Eq. (72) and $\beta$ is the inverse temperature already defined in Appendix F. From Eq. (73), it can be easily seen that the correlation function is periodic in $\varepsilon$: $F(k, \beta - \varepsilon) = F(k, \varepsilon)$. It is then necessary to sample $F(k, \varepsilon)$ only up to $\beta/2$, i.e. half of the polymer representing each quantum particle in PIMC terminology. With the PIMC simulation, we have sampled $F(k, \varepsilon)$ at the discrete points in which the action at temperature $T$ is decomposed. The initial point at $\varepsilon = 0$ corresponds to the zero energy-weighted sum rule $m_0$ of the dynamic response, which is the static structure factor $S(k)$ at that specific value of the wavenumber $k$

$$m_0 = S(k) = \int_{-\infty}^{+\infty} d\omega S(k, \omega).  \tag{74}$$

In order to carry out the inverse Laplace transform of the PIMC results of $F(k, \varepsilon)$, we need a reliable model for $S(k, \omega)$. We have chosen the step-wise function

$$S_m(k, \omega) = \sum_{i=1}^{N_s} \xi_i \Theta(\omega - \omega_i) \Theta(\omega_{i+1} - \omega),  \tag{75}$$

where $\Theta(x)$ is the Heaviside step function, and $\xi_i$ and $N_s$ are parameters of the model. Since the system under consideration, Eq. (2), is homogeneous and translationally invariant, the response functions depend only on the modulus $|k|$. By employing Eq. (75) in Eq. (73), we have obtained the corresponding model for the correlation function

$$F_m(k, \varepsilon) = \sum_{i=1}^{N_s} \xi_i \left\{ \frac{1}{\varepsilon} \left( e^{-\varepsilon \hbar \omega_i} - e^{-\varepsilon \hbar \omega_{i+1}} \right) + \frac{1}{\beta - \varepsilon} \left[ e^{-(\beta - \varepsilon) \hbar \omega_i} - e^{-(\beta - \varepsilon) \hbar \omega_{i+1}} \right] \right\}.  \tag{76}$$

Thanks to Eq. (76), the ill-conditioned character of the inverse Laplace transform has been converted into a multivariate optimization problem which tries to reproduce the PIMC data with the proposed model, Eq. (76). To this end, we have employed the simulated annealing method which relies on a thermodynamic equilibration procedure from high to low temperature according to a predefined template schedule [57]. The cost function which needs to be minimized is the quadratic dispersion

$$\chi^2(k) = \sum_{i=1}^{N_p} \left[ F(k, \varepsilon_i) - F_m(k, \varepsilon_i) \right]^2,  \tag{77}$$

where $N_p$ is the number of points in which the PIMC estimation of the $F(k, \varepsilon_i)$ is sampled. One may introduce the statistical errors coming from the PIMC simulations as the denominator of

Eq. (77). However, we have checked that this affects in a negligible way the final result since the size of the errors is rather independent of $\varepsilon$, see also Ref. [51].

The optimization leading to $S(k, \omega)$ has been carried out over a number $N_t$ of independent PIMC calculations of $F(k, \varepsilon)$. Typically, we have worked with $N_t = 20$ and for each one we have performed a number of $N_a = 100$ of independent simulated annealing searches. The mean average of these $N_a$ optimizations was our prediction for $S(k, \omega)$ for a given $F(k, \varepsilon)$.

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
