# Peer review of "Hole-induced anomaly in the thermodynamic behavior of a one-dimensional Bose gas"

_SciPost Physics, doi:SciPost Phys. 13, 035 (2022)_

## Round 3 · Referee Report · Anonymous (Referee 1) · 2022-3-21

Strengths

  • The authors report evidence of an anomaly in the thermodynamics of the Lieb-Lininger model describing one-dimensional Bose gases with contact repulsive interactions. The anomaly corresponds to a peak in the temperature dependence of the specific heat per unit length.

  • The anomaly temperature is found to be of the order of the energy of the maximum of the hole branch, suggesting a Schottky-like origin.

  • Path Integral Monte Carlo (PIMC) is used to extract the dynamic structure factor of the Bose gas as a function of temperature, allowing to access the changes in the excitation spectrum of the system leading to the anomaly.

  • The thermodynamics is explored through different analytical techniques, covering different regimes (wek interactions, strong interactions, high temperature ...)

Weaknesses

Some key-points of the manuscript remain unclear for a non expert.

-The identification of the anomaly temperature T_A from the behavior of the dynamic structure factor does not seem so evident.

  • The anomaly in the specific heat is very pronounced in the Tonks gas regime, whose thermodynamics is identical to that of the ideal Fermi gas. For weak interactions, say gamma=0.1 in Fig.1, the anomaly in the specific heat seems to be associated to a kink and not to a peak. This is confusing.

Report

The manuscript is rather interesting and contains new analytical and numerical results that deserve publication. However, for a non-expert, certain points remain obscur and should be clarified.

Some questions concern the interpretation of the dynamic structure factor results obtained from the PIMC presented in Fig.6:

  1. The authors say that the anomaly temperature T_A corresponds to a change of behavior of the dynamic structure factor from a low temperature quantum regime, controlled by interaction effects, and a high temperature thermal regime, where correlations are less relevant. Why I can say that around tau=0.173 there is a change of behavior ? The question is the same for Fig.10, maybe starting from the strong coupling regime the identification of T_A is easier ?

  2. If the temperature T_A is related to Delta, which is the maximum of the hole branch for k=k_F, why the authors use smaller values of k in Fig.6 ?

The author should also specify what is the definition of the anomaly, in particular for the weakly interacting regime corresponding to small gamma: is there a real maximum in the specific heat as for the chemical potential or is there just a kink ? This is particularly important to understand Fig.3: how the data for T_A are extracted and whether the anomaly exists for any nonzero gamma or only for gamma larger than a threshold.

REQUESTED CHANGES: The authors should address the point raised in the report. Some sentence are complicated to follow and should be made more explicit, for instance: "the breakdown of the quasiparticle picture, Eq. (12) is due to the thermal shift in the values of the frequency of the structure of the DSF".

Requested changes

The authors should address all the points raised in the report.
Some sentence are complicated to follow and should be reformulated for better clarity, for instance:
"the breakdown of the quasiparticle picture, Eq. (12) is due to the thermal shift in the values of the frequency of the structure of the DSF".

  • validity: high
  • significance: good
  • originality: high
  • clarity: ok
  • formatting: good
  • grammar: excellent

Author:  Giulia De Rosi  on 2022-05-24  [id 2511]

(in reply to Report 1 on 2022-03-21)

We thank Referee 1 for his/her genuine interest in our work and for his/her insightful suggestions and criticisms which have helped us to clarify some key issues and to greatly improve our manuscript.

1) The authors say that the anomaly temperature T_A corresponds to a change of behavior of the dynamic structure factor from a low temperature quantum regime, controlled by interaction effects, and a high temperature thermal regime, where correlations are less relevant. Why I can say that around tau=0.173 there is a change of behavior? The question is the same for Fig.10, maybe starting from the strong coupling regime the identification of T_A is easier?

The change of the behavior is signaled by a thermal broadening of the dynamic structure factor (DSF) as a function of frequency and for fixed values of the wavenumber. For low temperatures, the quasiparticle description holds, and for a given momentum only a very narrow range of frequencies is excited. Instead, for higher temperatures, the excitations are no longer exhausted by a single mode, so that modes within a wide range of frequencies are populated. Such a broadening of the peak becomes noticeable around the anomaly temperature T_A. For intermediate strength of interactions (gamma = 1) shown in Fig. 6, the anomaly temperature is located around tau = 0.173, and for strong interactions (gamma = 10) reported in Fig. 10 it is roughly tau = 0.699.

Differently to higher spatial dimensionalities, where a true phase transition occurs at the critical temperature, and thus the change of the DSF at the critical temperature is more evident, in one dimension (1D) there is instead a temperature crossover which makes smoother the broadening of the DSF peak. As a result, the anomaly temperature T_A found from the specific heat dependence, provides only an appropriate energy scale rather than a precise value at which the DSF broadening is observed.

The relationship between T_A and the appearance of thermal broadening in the DSF can be also seen in Figs. 5 (gamma = 1) and 9 (gamma = 10). In the first panels where T< T_A, the density plot of the DSF can be well approximated by the single-mode quasiparticle description, Eq. (12), for any k. In the second panel where the temperature is around T_A, there is instead the breakdown of the quasiparticle approximation.

It is important to notice that results at different gamma (comparing Fig. 6 with 10 and Fig. 5 with 9) are qualitatively similar, showing that the thermal broadening of the DSF at T_A discussed above is a universal phenomenon with the interaction strength, as claimed in Sec. 5. This can be also understood by recalling that there is a crossover by keeping fixed the temperature and changing gamma, as discussed in Sec. 2.

The Referee has raised a crucial point which was not entirely clear in the manuscript. We added additional text at the end of Sec. 5. We thank the Referee for the very insightful comment.

2) If the temperature T_A is related to Delta, which is the maximum of the hole branch for k=k_F, why the authors use smaller values of k in Fig.6 ?

The Path-Integral Monte Carlo method allows access to imaginary-time correlations, which are used to extract the frequency dependence of the Dynamic Structure Factor by performing the Inverse Laplace method. Being a mathematically ill-posed problem, the quality of the inverse Laplace transform depends on the momentum k and interaction parameter gamma.

While the results for k = k_F are already contained in the density plot of Fig. 5, where the peak of the DSF is broad, its signal is low and the resolution of the Inverse Laplace transform is limited. We have tried to build the corresponding of Fig. 6 at k = k_F, but since the DSF peak is broad even at small temperatures, the change of the DSF at the anomaly is even less evident and we prefer to not add the new plot, also it is redundant with Fig. 5.

In Fig. 10 where gamma = 10, we have indeed reported results for k = k_F but they are not qualitatively different from those at smaller k or gamma = 1 of Fig. 6.

3) The author should also specify what is the definition of the anomaly, in particular for the weakly interacting regime corresponding to small gamma: is there a real maximum in the specific heat as for the chemical potential or is there just a kink? This is particularly important to understand Fig.3: how the data for T_A are extracted and whether the anomaly exists for any nonzero gamma or only for gamma larger than a threshold.

The Referee is certainly correct when says that for weak interactions, the anomaly temperature does not correspond to a maximum in the specific heat. Enclosed to the current reply, we present the first derivative of the specific heat with respect to temperature for

small values of gamma: https://www.dropbox.com/s/9d6j63jmi3sz3d5/DCSmallGamma.pdf?dl=0

large values of gamma: https://www.dropbox.com/s/s3zpu7mqk06y57x/DCLargeGamma.pdf?dl=0

For both plots, the vertical lines correspond to the anomaly temperature reported in Fig. 3.

For gamma >= 1, the position of the thermal feature (i.e. anomaly) in the specific heat corresponds to its local maximum. The anomaly temperature T_A (see Fig. 3) has been evaluated as the point where the first derivative of the specific heat is zero (see new plot). Please notice that for gamma = 1, the anomaly corresponds to the zero of the derivative at the lowest temperature.

For gamma << 1, the position of the anomaly in the specific heat cannot be anymore interpreted as a maximum, as the specific heat is a monotonically increasing function with temperature, thus T_A cannot be defined then by the zero first derivative. While the anomaly resembles a kink in this case, mathematically speaking it is not, as a kink would correspond to a discontinuous first derivative of the specific heat. Instead, in the 1D case the energy and the specific heat are both continuous functions. Still, the first derivative experiences a change around the anomalous temperature. In this case, T_A has been extracted as the temperature at which the first derivative of the specific heat experiences a significant change of its first derivative (see new plot at small gamma). By decreasing gamma, the anomaly reduces as the corresponding hole energy Delta gets smaller, see Fig. 3. Our description for the anomaly is valid for any gamma and it is also reflected by the same qualitative behavior of the DSF at T_A along the interaction crossover, as discussed above in the current reply.

We have introduced small changes throughout the manuscript, where we define the anomaly as a thermal feature in the specific heat. We really thank the Referee for the useful suggestion.

4) Some sentence are complicated to follow and should be made more explicit, for instance: "the breakdown of the quasiparticle picture, Eq. (12) is due to the thermal shift in the values of the frequency of the structure of the DSF".

We agree with the Referee that this sentence might be complicated to follow. We rephrased it by relating the breakdown of the quasiparticle picture with the thermal broadening of the peak of the DSF, in agreement with the above discussion. Changes to the text have been implemented in Secs. 1, 5 and 7. We thank the Referee for the insightful observation.

---

## Round 3 · Referee Report · Anonymous (Referee 2) · 2022-4-4

Report

The authors consider the finite temperature specific heat at constant density in the repulsive one-dimensional Lieb-Liniger model. As the model is integrable this can be straightforwardly done by numerically solving the well-known thermodynamic Bethe Ansatz equations. The authors note that the specific heat displays a non-monotonic behavior as a function of temperature and this allows them to identify a temperature scale T_A. They go on to relate this scale to the gap of the hole branch in the excitation spectrum at momentum k_F. To substantiate this relation the dynamical structure factor is determined by path integral Monte-Carlo methods.

While this manuscript contains a lot of useful material, I am not convinced that it fulfils the stringent acceptance criteria for SciPost Physics.

  1. The main message of this work is that the specific heat at constant density and other quantities like the chemical potential exhibit maxima at a finite temperature. I do not find this surprising. 1D quantum spin-chains are well-known to display such behavior and it is easy to understand by considering both the small and large-temperature limits. The maximum occurs at a temperature of order of the exchange energy. I note that the authors comment on the situation for quantum spin chains themselves. With regards to quantum gases, the free Fermi gas also displays this behavior, which can be seen by a straightforward analysis. In my understanding this means that the scale identified by the authors must necessarily have a (kinematic) origin.

  2. While I find the results obtained by the various approximation schemes shown in Fig 1 useful from a general perspective, they arguably add nothing substantial to the topic of the paper, i.e. the "hole anomaly". Some readers might feel that these kinds of comparisons would be more appropriate for a review article or text book on Bose gases.

  3. As is clear from Fig.3 T_A and \Delta are not precisely the same, but are only of the same order. Similarly, the maximum in the chemical potential is not quite the same as T_A. This is perhaps as expected, and indicates the presence of one characteristic energy scale that gives rise to T_A, \Delta etc in a non-universal way that depends on the details of the quantity that one considers. This then poses the question whether there is anything physically significant about the fact that T_A is similar to \Delta.

  4. There are a number of statements in the Conclusions that I find difficult to follow.

(i) Our description for the hole anomaly is valid for any value of the interaction strength γ and solves the open problem of relating the effects of the complicated continuous spectrum to the thermodynamic behavior of a 1D Bose gas.

I don't understand this claim.

(ii) The anomaly is a reminiscence of a phase transition, that is not allowed in 1D systems, and signals a change of quantum regime.

I disagree: I do not think there are is any similarity with the "anomaly" and a phase transition.

(iii) A tantalizing possibility is that the hole anomaly could be employed as a quantum simulator as it provides an in-depth understanding of diverse anomalies in other more complicated many-body systems. The new anomaly shares typical properties of a quantum simulator as its future observation is feasible and it can be achieved in clean experimental atomic settings where precise control and broad tunability of the interaction strength γ are possible.

I am not sure I understand what is meant by this (other than the effect should be measurable in experiments).

  • validity: high
  • significance: ok
  • originality: ok
  • clarity: ok
  • formatting: excellent
  • grammar: excellent

Author:  Giulia De Rosi  on 2022-05-24  [id 2512]

(in reply to Report 2 on 2022-04-04)

We thank Referee 2 for his/her genuine interest in our work and for his/her insightful suggestions and criticisms which have helped us to clarify some key issues and to greatly improve our manuscript.

1) The main message of this work is that the specific heat at constant density and other quantities like the chemical potential exhibit maxima at a finite temperature. I do not find this surprising. 1D quantum spin-chains are well-known to display such behavior and it is easy to understand by considering both the small and large-temperature limits. The maximum occurs at a temperature of order of the exchange energy. I note that the authors comment on the situation for quantum spin chains themselves. With regards to quantum gases, the free Fermi gas also displays this behavior, which can be seen by a straightforward analysis. In my understanding this means that the scale identified by the authors must necessarily have a (kinematic) origin.

The main message of our work is providing the physical interpretation for the presence of the anomaly (i.e a thermal feature) in the temperature dependence of the specific heat rather than its existence. The presence of the maximum in the chemical potential is known from the thermal Bethe-Ansatz (TBA) Yang-Yang solution. However, to the best of our knowledge the physics of the specific heat in 1D continuum models has not been never addressed before. The chemical potential has been already evaluated by 2 of the Authors with TBA in Ref. [19] and [33] (as discussed at the end of Sec. 4.1.), for the whole interaction crossover and from the quantum to the classical regime. The analytical limit of the one-dimensional (1D) ideal Fermi gas has been calculated as well in the same previous works.

However, the TBA does not provide any physical explanation for the anomaly. “The investigation of the thermodynamic properties of the one-dimensional Bose gas is a difficult problem. Yang and Yang (1969) derived exact equations which allow the calculation [...] at arbitrary temperatures. However, it is not easy to give a simple interpretation of these equations in terms of elementary excitations” (see pag. 500-502 of the book Ref. [28]). Our work fills this important gap by interpreting the anomaly in thermodynamic properties in terms of the maximal hole energy Delta in the excitation spectrum and a change in the behavior of the Dynamic Structure Factor (DSF). This is the main message of our work, which is repeated in the Abstract, Secs. 1, 4, 5, and 7.

It is an interesting question if the energy scale defining the position of the anomaly in the thermodynamic functions has a kinematic origin. Definitely, in the Tonks-Girardeau (TG) regime at zero temperature, the only relevant scale is the Fermi energy of an ideal Fermi gas (IFG). At finite temperature, since interactions are not present in the IFG model, only the kinetic energy (induced by the temperature) plays a role in the deep TG gas. Instead in the mean-field limit, there is also the contribution of the interaction energy. Thus, depending on the regime, the interaction between atoms might be dominant or negligible. The scale for the anomaly temperature is provided by maximal hole energy Delta which is a function of the interaction strength gamma, see Eq. (1). In addition, as discussed in the Abstract, Secs. 1, 4, 5 and 7, the anomaly originates from the balance between the interaction effects (encoded in Delta) and the temperature (i.e. kinetic motion of atoms).

2) While I find the results obtained by the various approximation schemes shown in Fig 1 useful from a general perspective, they arguably add nothing substantial to the topic of the paper, i.e. the "hole anomaly". Some readers might feel that these kinds of comparisons would be more appropriate for a review article or text book on Bose gases.

We wish to keep the limiting approximations reported in Fig. 1 for a number of reasons:

*Fig. 1 can be more easily interpreted as a diagram of different regimes

*It is clear that the hole anomaly cannot be described by the approximated theories of Sec. 3 and it instead requires exact methods such as TBA and Path Integral Monte Carlo (PIMC)

*The anomaly separates different regimes. It is important to notice that some of the theoretical approximations have been employed to get limits at small and large gamma for the anomaly temperature reported in Fig. 3.

*As also noticed by the other 2 Referees, this is a strength of the paper. We strongly believe that the various approximations make our work more complete.

We added a comment to Sec. 3 clarifying that the presence of the various limits makes it easier to interpret Fig. 1 as a diagram of regimes. We thank the Referee for the point raised.

3) As is clear from Fig.3 T_A and \Delta are not precisely the same, but are only of the same order. Similarly, the maximum in the chemical potential is not quite the same as T_A. This is perhaps as expected, and indicates the presence of one characteristic energy scale that gives rise to T_A, \Delta etc in a non-universal way that depends on the details of the quantity that one considers. This then poses the question whether there is anything physically significant about the fact that T_A is similar to \Delta.

In Eq. (1), we indeed claimed that T_A and \Delta are of the same order (and not necessarily equal). Our qualitative explanation of the hole anomaly is actually based on Eq.(1) and on the mapping with the well-known Schottky Anomaly in the two-level model where T_A is only of the same order as Delta: T_A = 0.4 \Delta (see Sec. 1). In the revised version of the manuscript, we added a comment in Sec. 7 discussing the important analogies between our hole anomaly in the 1D Bose gas and other anomalies in spin systems. It is worth noticing that even in these latter systems, the anomaly temperature is only of the same order of the spin energy gap in the spectrum playing the same role of our Delta.

The position of the maximum (i.e. anomaly temperature) in the chemical potential is different from the anomaly temperature of the specific heat (even if they are of the same order). Even in spin systems, the anomaly temperatures in different thermodynamic quantities such as the specific heat and the magnetization are not precisely the same, although they are of the same order, see new comment in Sec. 7.

The discrepancy between the anomaly temperature and the energy Delta as well as the differences between the anomaly temperatures of various thermodynamic properties ​​are expected, provided they are of the same order of magnitude. This intriguing feature is universal of all kinds of anomalies occurring in very different systems.

4) There are a number of statements in the Conclusions that I find difficult to follow.

(i) Our description for the hole anomaly is valid for any value of the interaction strength γ and solves the open problem of relating the effects of the complicated continuous spectrum to the thermodynamic behavior of a 1D Bose gas.

We find that the hole anomaly is present for any finite interaction strength gamma. This is clear from Eq.(1), the peak in the specific heat in Fig. 1 and the DSF (by comparing results in Sec. 5 and Appendix G for different gamma).

As also discussed in the first point of this reply, the novel anomaly allowed us to solve the important open problem of the understanding of the behavior of the thermodynamic quantities in terms of the complicated structure of the excitation spectrum.

The Referee is right when says that this point of the manuscript may result unclear. We have deleted “continuous” when referred to the spectrum in Secs. 1, 4 and 7. We thank the Referee for the useful comment.

(ii) The anomaly is a reminiscence of a phase transition, that is not allowed in 1D systems, and signals a change of quantum regime.

I disagree: I do not think there are is any similarity with the "anomaly" and a phase transition.

As explained in Sec. 1, any thermal second-order phase transition exhibits an anomaly in the specific heat located at the critical temperature T_c. As discussed at the end of Sec. 5, the DSF at the anomaly temperature T_A shows a similar behavior as in superfluid phase transitions at T_c in Bose systems of higher spatial dimensionalities. There is then a clear mapping between T_A vs T_c and the anomaly vs phase transitions.

(iii) A tantalizing possibility is that the hole anomaly could be employed as a quantum simulator as it provides an in-depth understanding of diverse anomalies in other more complicated many-body systems. The new anomaly shares typical properties of a quantum simulator as its future observation is feasible and it can be achieved in clean experimental atomic settings where precise control and broad tunability of the interaction strength γ are possible.

I am not sure I understand what is meant by this (other than the effect should be measurable in experiments).

This paragraph can be understood if completed with the next 2 sentences, until “quantum technologies”. Here we claim that the hole anomaly satisfies all the properties of a quantum simulator which are listed in the same part of the manuscript. So that, it can be employed as a model of other anomalies in different solid-state, electronic and spin systems.

An important property of being a quantum simulator is the experimental realization where all parameters can be changed at will. The interaction strength gamma ruling Eq.(1) of the anomaly can be tuned with the well-established Fano-Feshbach resonances experimental technique (see Sec. 7).

We are convinced that the hole anomaly can be observed in the near future at least (see Sec. 7):

*Through the behavior of the DSF which was already measured but at T < T_A

*Through the chemical potential already measured for a broad range of temperatures including the anomaly

In the point raised by the Referee, we have listed the kinds of many-body systems whose anomalies may be simulated by our hole anomaly. In Sec. 7, we added an additional comment about the measurement of the chemical potential. We thank the Referee for the useful observation.

---

## Round 3 · Referee Report · Anonymous (Referee 3) · 2022-4-19

Strengths

1) An interesting study of the physical properties of quantum interacting bosons, a subject central to both cold atoms and quantum spin systems 2) a combined analytical and numerical study giving physical hindsight on the physics of the system 3) a well written paper

Weaknesses

1) some absence of contact with previous literature on the same topic in the context of spin systems (both for specific heat and magnetization curves) 2) Some additional informations needed to fully evaluate the possibilities offered by the excellent calculation of the dynamical structure factor performed using QMC in the present paper.

Report

This paper studies the physical properties, in particular the specific heat and dynamical structure factor, of a one dimensional gas of bosons with a contact interaction (Lieb-Lininger model).

Using a combination of analytical (Bethe-ansatz (BA) and various approximate techniques) and numerical (Quantum monte-carlo (QMC)) the authors investigate the existence of a maximum in the temperature dependence of the specific heat, beyond the linear regime described by the Tomonaga-Luttinger liquid (TLL) physics. The connect this maximum to the excitation spectrum. They also compute numerically the dynamical structure factor and compare it with the BA and other analytical results.

The question of the physical properties of a one dimensional gas of interacting bosons is of course central in many systems ranging from cold atoms to quantum spin chains. Understanding these physical properties in a reliable way is thus of great importance. The low energy properties are very well described by effective field theories such as the TLL, but going beyond is difficult beyond the precise but often opaque solutions given by integrable models (BA).

I thus find the study performed by the authors, namely: i) the detailed analysis of the peak in the specific heat and its connection to the Lieb spectrum. ii) the comparison with the various approximate analytical formulas at high and low temperatures. iii) the use of quantum monte-carlo to obtain the dynamical structure factor (with the delicate problem of the analytic continuation) both new and interesting.

In addition the paper is well written and there is thus no doubt that the paper is suitable for publication in Scipost Physics.

I have however a certain number of remarks and questions that must be addressed by the authors prior to publication:

  • the question of the peak in specific heat, was already addressed for hard core bosons (spinless fermions) in the context of quantum spin chains and ladders. In that context the end of the TLL linear regime was naturally interpreted as the moment when the thermal energy coincides with the Fermi energy of the spinless fermions thus allowing excitations to reach the bottom of the band rather than the linearized regions around the Fermi points, thus changing in effect the spectrum from linear to k^2. Together with the effects in the specific heat, this also coincides with a minimum in the magnetization vs temperature curve (in the language of bosons: density versus temperature). For example the authors can look at : Y. Maeda et al. PRL 99, 057205 (2007); C. Ruegg et al. PRL 101 247202 (2008); P. Bouillot et al. PRB 83, 054407 (2011). For the hard core limit this seems to coincide with the criterion used by the authors in the present paper.

It would thus be interesting that the more complete and detailed analysis made by the authors is connected to this previous body of works. In particular to explore the connection with the density vs temperature curve for the soft core bosons as well.

  • Besides BA several calculations of the structure factor have been performed for bosons using Density Matrix Renormalization Group (DMRG) methods as well (see e.g. the last reference above and references therein for the case of quantum spin systems). Of course the DMRG is well suited for lattice systems, and when the number of states per sites is not too large. It has however the advantage to allow direct calculation in real time. I thus find the numerical study of the structure factor performed using QMC particularly interesting since it allows to address a priori the continuous limit. However as pointed out by the authors the calculation is done in imaginary times and the question of analytic continuation is central. The authors seem to have obtained very good results shown e.g. in Fig~9 and explained in part their analytic continuation method in section 6) and have benchmarked some of the QMC results against BA in the appendix.

Given the interest of obtaining the dynamical structure factors it could however be interesting to have slightly more detailed on the analytical continuation procedure, and perhaps a comparison with the free spinless fermion analytical results which is easy to obtain and give the structure factor at $\gamma = \infty$.

With these points taken into consideration by the authors the paper should be suitable for publication.

Requested changes

The authors should consider and address the points mentioned above. No mandatory changes are requested.

  • validity: high
  • significance: high
  • originality: high
  • clarity: high
  • formatting: excellent
  • grammar: excellent

Author:  Giulia De Rosi  on 2022-05-24  [id 2513]

(in reply to Report 3 on 2022-04-19)

We thank Referee 3 for his/her genuine interest in our work and for his/her insightful suggestions and criticisms which have helped us to clarify some key issues and to greatly improve our manuscript.

1) the question of the peak in specific heat, was already addressed for hard core bosons (spinless fermions) in the context of quantum spin chains and ladders. In that context the end of the Tomonaga-Luttinger liquid (TLL) linear regime was naturally interpreted as the moment when the thermal energy coincides with the Fermi energy of the spinless fermions thus allowing excitations to reach the bottom of the band rather than the linearized regions around the Fermi points, thus changing in effect the spectrum from linear to k^2. Together with the effects in the specific heat, this also coincides with a minimum in the magnetization vs temperature curve (in the language of bosons: density versus temperature). For example the authors can look at : Y. Maeda et al. PRL 99, 057205 (2007); C. Ruegg et al. PRL 101 247202 (2008); P. Bouillot et al. PRB 83, 054407 (2011). For the hard core limit this seems to coincide with the criterion used by the authors in the present paper.

It would thus be interesting that the more complete and detailed analysis made by the authors is connected to this previous body of works. In particular to explore the connection with the density vs temperature curve for the soft core bosons as well.

In Sec. 4.1, we recover Eq.(10) which is equivalent to the anomaly relation Eq.(1), by considering the scale of the upper bound of the validity of the TLL regime (called in our work Luttinger Liquid) which has been estimated by comparing the chemical potential at zero temperature with the TLL-thermal correction (depending on the square of temperature T). Eq.(10) depends on the interaction strength gamma through the sound velocity v, so it is valid in the entire crossover from soft- to hard-core bosons.

In the limit of the Tonks-Girardeau (TG) regime where deeply hard-core bosons behave as spinless fermions, v is equal to the Fermi value (see Sec. 2) and Eq.(10) exactly recovers that the anomaly occurs when the thermal energy is comparable to the Fermi scale. As the Referee pointed out, the same also takes place in spin chains and ladders in the regime of spinless fermions. Our results are then more general as holding in the entire interaction crossover. The Referee indeed helped us a lot to identify other analogies with spin chains and ladders. This makes the concept of quantum simulation of the novel hole anomaly even stronger and broader than what we expected before.

We added a new paragraph in Sec. 7 discussing the important analogies with spin chains and ladders with connections to the insightful works cited in this Report 3. We really thank the Referee for this crucial observation.

The density vs temperature plot is equivalent to the gamma vs T diagram of the different regimes and contains the same information of Fig. 1. Different regimes can be easily identified by various approximated limits of Sec. 3. Thermal Bethe-Ansatz (TBA) and Path Integral Monte Carlo (PIMC) results have been obtained with fixed values of gamma. It would be difficult to perform the same exact calculations for infinite discretized values of gamma such that it can be approximated as a continuous quantity in order to build the gamma vs T diagram which is somehow redundant with Fig. 1.

2) Besides BA several calculations of the structure factor have been performed for bosons using Density Matrix Renormalization Group (DMRG) methods as well (see e.g. the last reference above and references therein for the case of quantum spin systems). Of course the DMRG is well suited for lattice systems, and when the number of states per sites is not too large. It has however the advantage to allow direct calculation in real time. I thus find the numerical study of the structure factor performed using QMC particularly interesting since it allows to address a priori the continuous limit. However as pointed out by the authors the calculation is done in imaginary times and the question of analytic continuation is central. The authors seem to have obtained very good results shown e.g. in Fig~9 and explained in part their analytic continuation method in section 6) and have benchmarked some of the QMC results against BA in the appendix.

Given the interest of obtaining the dynamical structure factors it could however be interesting to have slightly more detailed on the analytical continuation procedure, and perhaps a comparison with the free spinless fermion analytical results which is easy to obtain and give the structure factor at gamma = infinite.

We thank the Referee for the recognition that the calculation of the dynamic structure factor (DSF) is an interesting but as well very complicated question. We have tried to calculate the DSF from the Inverse Laplace transform of PIMC data in the deep hard-core regime with gamma = 100, which should reproduce the analytical result of the TG limit where gamma = infinity, see Fig. 3. However, the diverging interaction potential makes the numerical calculations in this TG limit complicated. It is important to say that the analytical DSF in the TG regime has been already obtained in Ref.[26] cited in Sec. 1.

We agree with the Referee on the importance of the question of the analytic continuation (called simulated annealing in our work). We have then added a detailed appendix describing such a procedure. Following this suggestion, we have also added a new comment in Sec. 7, by comparing the features of DMRG and PIMC methods with the inclusion of new references. We really thank the Referee for the important suggestion.

---

## Round 4 · Referee Report · Anonymous (Referee 2) · 2022-6-7

Report

The authors have not really addressed many of my concerns in their revised version. Most importantly they have not addressed my comment 1), namely that as the free Fermi gas displays the same behavior discussed in the manuscript the scale identified by the authors should simply have a kinematic origin. I also find their reply to my comment 4) about the claimed similarities between the "anomaly" and a phase transition very strange: the free Fermi gas displays this behavior and surely does not have a phase transition.

I do not think that the revised version satisfies the stringent publication criteria of SciPost Physics, but it would be suitable for SciPost Physics Core.

---

## Round 4 · Referee Report · Anonymous (Referee 1) · 2022-6-8

Strengths

  • An interesting study of the properties of a 1D integrable quantum model at finite temperature.

  • A combination of different techniques to compute key observables, including the Bethe ansatz (specific heat) and the path-integral Monte Carlo (dynamic structure factor).

  • A physical interpretation of the anomaly in the specific heat for the Lieb-Lininger model.

Report

The authors have provided a detailed and honest answer to all my remarks and criticisms. The presentation of the results has also been significantly improved, including a discussion on the procedure used to extract the anomaly temperature T_A from the specific heat numerical data.
The manuscript contains new and interesting results of one-dimensional integrable models at finite temperature. I therefore recommend publication in SciPost Physics.

---

## Round 4 · Referee Report · Anonymous (Referee 3) · 2022-6-23

Report

The modifications made by the authors have addressed my previous comments. In particular there is now a satisfactory discussion of the observed anomaly in a broader context, connecting also the calculations to more intuitive explanations. The discussion on the monte-carlo part has also been improved.

In connection with my previous comments on the scientific qualities of the papers and the results, I think that the current version is now perfectly adapted for publication in SciPost Physics and I do recommend publication in its present form.

---

## Round 4 · List of Changes

Changes to the text are marked in the file:
https://www.dropbox.com/s/ldhwt66umouaur7/diff.pdf?dl=0

A list of main changes follows:

1) New text close to the end of Sec. 5 on the change of the behavior of the Dynamic Structure Factor (DSF) upon request of Report 1.

2) Small changes to the text about the definition of anomaly in Secs. 1, 4.1, 7 and caption of Fig. 3 upon request of Report 1.

3) Changes to the text related to the breakdown of the quasiparticle picture in Secs. 1, 5 and 7 upon the suggestion of Report 1.

4) New comment in Sec. 3 about the importance of approximated limits of the specific heat for the interpretation of Fig. 1 as a diagram of regimes following the point raised in Report 2.

5) We have deleted “continuous” when referred to the spectrum in Secs. 1, 4, and 7 following the point raised in Report 2.

6) We have listed the kinds of many-body systems whose anomalies may be simulated by our hole anomaly, after the comment in Report 2.

7) New sentence about the measurement of the chemical potential for temperatures below and above the hole anomaly in Sec. 7 following the comment of Report 2.

8) New paragraph about spin chains and ladders in Sec. 7 with the inclusion of references suggested by Referee 3. New reference to spin ladders in the Abstract, upon the great suggestion of Report 3.

9) Change of the Acknowledgements Section, with the inclusion of Referee 3.

10) New Appendix about the simulated annealing which is the method employed for the calculation of the DSF and reference to it at the end of Sec. 6 upon request of Report 3.

11) New comment in Sec. 7 with the comparison of DMRG and PIMC techniques, upon request of Report 3.

---

## Editorial Decision

published